# Acceleration with a Ball Optimization Oracle

**Yair Carmon**[*†]  **Arun Jambulapati**[*]  **Qijia Jiang**[*]  **Yujia Jin**[*]  **Yin Tat Lee**[‡]

**Aaron Sidford**[*]  **Kevin Tian**[*]

## Abstract

Consider an oracle which takes a point $x$ and returns the minimizer of a convex function $f$ in an $\ell_2$ ball of radius $r$ around $x$. It is straightforward to show that roughly $r^{-1} \log \frac{1}{\epsilon}$ calls to the oracle suffice to find an $\epsilon$-approximate minimizer of $f$ in an $\ell_2$ unit ball. Perhaps surprisingly, this is not optimal: we design an accelerated algorithm which attains an $\epsilon$-approximate minimizer with roughly $r^{-2/3} \log \frac{1}{\epsilon}$ oracle queries, and give a matching lower bound. Further, we implement ball optimization oracles for functions with locally stable Hessians using a variant of Newton's method and, in certain cases, stochastic first-order methods. The resulting algorithm applies to a number of problems of practical and theoretical import, improving upon previous results for logistic and $\ell_\infty$ regression and achieving guarantees comparable to the state-of-the-art for $\ell_p$ regression.

## 1 Introduction

We study unconstrained minimization of a smooth convex objective $f : \mathbb{R}^d \to \mathbb{R}$, which we access through a *ball optimization oracle* $\mathcal{O}_{\text{ball}}$, that when queried at any point $x$, returns the minimizer of $f$ restricted a ball of radius $r$ around $x$, i.e.,[1]

$$\mathcal{O}_{\text{ball}}(x) = \underset{x' \text{ s.t. } \|x'-x\| \le r}{\arg\min} f(x').$$

Such oracles underlie trust-region methods [15] and, as we demonstrate via applications, encapsulate problems with local stability. Iterating $x_{k+1} \leftarrow \mathcal{O}_{\text{ball}}(x_k)$ minimizes $f$ in $\widetilde{O}(R/r)$ iterations (see Appendix A), where $R$ is the initial distance to the minimizer, $x^*$, and $\widetilde{O}(\cdot)$ hides polylogarithmic factors in problem parameters, including the desired accuracy.

Given the fundamental geometric nature of the ball optimization abstraction, the central question motivating our work is whether it is possible to improve upon this $\widetilde{O}(R/r)$ query complexity. It is natural to conjecture that the answer is negative: we require $R/r$ oracle calls to observe the entire line from $x_0$ to the optimum, and therefore finding a solution using less queries would require jumping into completely unobserved regions. Nevertheless, we prove that the optimal query complexity scales as $(R/r)^{2/3}$. This result has positive implications for the complexity for several key regression tasks, for which we can efficiently implement the ball optimization oracles.

### 1.1 Our contributions

We overview our main contributions: accelerating ball optimization oracles (with a matching lower bound), implementing them under Hessian stability, and applying our results to regression problems.

---

[*]Stanford University, {yairc,jmblpati,qjiang2,yujiajin,sidford,kjtian}@stanford.edu.

[†]Tel Aviv University, ycarmon@cs.tau.ac.il.

[‡]University of Washington, yintat@uw.edu.

[1]In the introduction we discuss exact oracles for simplicity, but our results account for inexactness. Our results hold for any weighted Euclidean (semi)norm, i.e., $\|x\| = \sqrt{x^\top \mathbf{M} x}$ for $\mathbf{M} \succeq 0$, which we sometimes write explicitly as $\|x\|_{\mathbf{M}}$.

**Monteiro-Svaiter (MS) oracles via ball optimization.** Our starting point is an acceleration framework due to Monteiro and Svaiter [25]. It relies on access to an oracle that when queried with $x, v \in \mathbb{R}^d$ and $A > 0$, returns points $x_+, y \in \mathbb{R}^d$ and $\lambda > 0$ such that

$$y = \frac{A}{A + a_\lambda}x + \frac{a_\lambda}{A + a_\lambda}v, \text{ and} \tag{1}$$

$$x_+ \approx \underset{x' \in \mathbb{R}^d}{\arg\min}\left\{f(x') + \frac{1}{2\lambda}\|x' - y\|^2\right\}, \tag{2}$$

where $a_\lambda = \frac{1}{2}(\lambda + \sqrt{\lambda^2 + 4A\lambda})$. Basic calculus shows that for any $z$, the radius-$r$ oracle response $\mathcal{O}_{\text{ball}}(z)$ solves the proximal point problem (2) for $y = z$ and some $\lambda = \lambda_r^\star(z) \geq 0$ which depends on $r$ and $z$. Therefore, to implement the MS oracle with a ball optimization oracle, given query $(x, v, A)$ we need to find $\lambda$ that solves the implicit equation $\lambda = \lambda_r^\star(y(\lambda))$, with $y(\lambda)$ as in (1). We solve this equation to sufficient accuracy via binary search over $\lambda$, resulting in an accelerated scheme that makes $\widetilde{O}(1)$ queries to $\mathcal{O}_{\text{ball}}(\cdot)$ per iteration (each iteration also requires a gradient evaluation).

The main challenge lies in proving that our MS oracle implementation guarantees rapid convergence. We do so by a careful analysis which relates convergence to the distance between the MS oracle outputs $y$ and $x_+$. Specifically, letting $\{y_k, x_{k+1}\}$ be the sequence of these points, we prove that

$$\frac{f(x_K) - f(x^*)}{f(x_0) - f(x^*)} \leq \exp\left\{-\Omega(K)\min_{k<K}\frac{\|x_{k+1} - y_k\|^{2/3}}{R^{2/3}}\right\}.$$

Since $\mathcal{O}_{\text{ball}}$ guarantees $\|x_{k+1} - y_k\| = r$ for all $k$ except possibly the last, our result follows.

**Matching lower bound.** We give a distribution over functions with domain of size $R$ for which any algorithm interacting with a ball optimization oracle of radius $r$ requires $\Omega((R/r)^{2/3})$ queries to find an approximate solution with $O(r^{1/3})$ additive error. Our lower bound in fact holds for an even more powerful $r$-local oracle, which reveals all values of $f$ in a ball of radius $r$ around the query point. We prove our lower bounds using well-established techniques and Nemirovski's function, a canonical hard instance in convex optimization [26, 30, 12, 17, 9]. Here, our primary contribution is to show that appropriately scaling this construction makes it hard even against $r$-local oracles with a fixed radius $r$, as opposed to the more standard notion of local oracles that reveal the instance only in an arbitrarily small neighborhood around the query.

**Ball optimization oracle implementation.** Trust-region methods [15] solve a sequence of subproblems of the form

$$\underset{\delta \in \mathbb{R}^d \text{ s.t. } \|\delta\| \leq r}{\text{minimize}}\left\{\delta^\top g + \frac{1}{2}\delta^\top \mathbf{H}\delta\right\}.$$

When $g = \nabla f(x)$ and $\mathbf{H} = \nabla^2 f(x)$, the trust-region subproblem minimizes a second-order Taylor expansion of $f$ around $x$, implementing an approximate ball optimization oracle. We show how to implement a ball optimization oracle for $f$ to high accuracy for functions satisfying a local Hessian stability property. Specifically, we use a notion of *Hessian stability* similar to that of Karimireddy et al. [22], requiring $\frac{1}{c}\nabla^2 f(x) \preceq \nabla^2 f(y) \preceq c\nabla^2 f(x)$ for every $y$ in a ball of radius $r$ around $x$ for some $c > 1$. We analyze Nesterov's accelerated gradient method in a Euclidean norm weighted by the Hessian at $x$, which we can also view as accelerated Newton steps, and show that it implements the oracle in $\widetilde{O}(c)$ linear system solutions, improving upon the $c^2$ dependence of more naive methods. This improvement is not necessary for our applications where we take $c$ to be a constant, but we include it for completeness. For certain objectives (e.g., softmax), we show that a *first-order* oracle implementation (e.g., computing the Newton steps with accelerated SVRG) allows us to further exploit the problem structure, and improve state-of-the-art runtimes guarantees in some regimes.

**Applications.** We apply our implementation and acceleration of ball optimization oracles to problems of the form $f(\mathbf{A}x - b)$ for data matrix $\mathbf{A} \in \mathbb{R}^{n \times d}$. For logistic regression, where $f(z) = \sum_{i \in [n]} \log(1 + e^{-z_i})$, Hessian stability implies [4] that our algorithm solves the problem with $\widetilde{O}(\|x_0 - x^*\|_{\mathbf{A}^\top \mathbf{A}}^{2/3})$ linear system solves of the form $\mathbf{A}^\top \mathbf{D}\mathbf{A}x = z$ for diagonal $\mathbf{D}$. This improves upon the previous best linearly-convergent condition-free algorithm due to Karimireddy et al. [22], which requires $\widetilde{O}(\|x_0 - x^*\|_{\mathbf{A}^\top \mathbf{A}})$ system solves. Our improvement is precisely the power $2/3$ factor that comes from acceleration using the ball optimization oracle.

For $\ell_\infty$ regression, we take $f$ to be the log-sum-exp (softmax) function and establish that it too has a stable Hessian. By appropriately scaling softmax to approximate $\ell_\infty$ regression to $\epsilon$ additive

error and taking $r = \epsilon$, our method solves $\ell_\infty$ to additive error $\epsilon$ in $\widetilde{O}(\|x_0 - x^*\|_{\mathbf{A}^\top \mathbf{A}}^{2/3} \epsilon^{-2/3})$ linear system solves of the same form as above. This improves upon the algorithm of Bullins and Peng [11] in terms of $\epsilon$ scaling (from $\epsilon^{-4/5}$ to $\epsilon^{-2/3}$) and the algorithm of Ene and Vladu [18] in terms of distance scaling (from $n^{1/3}\|\mathbf{A}(x_0 - x^*)\|_\infty^{2/3}$ to $\|\mathbf{A}(x_0 - x^*)\|_2^{2/3}$). We also give a runtime guarantee improving over the state-of-the-art first-order method of Carmon et al. [13] whenever $\frac{n}{d} \geq (\frac{\max_i \|a_i\|_2 R}{\epsilon})^{2/3} \geq d$ where $R$ is the $\ell_2$ distance between an initial point and the optimizer, by using a first-order oracle implementation based on [5].

Finally, we leverage our framework to obtain high accuracy solutions to $\ell_p$ norm regression, where $f(z) = \sum_{i \in [n]} |z_i|^p$, via minimizing a sequence of proximal problems with geometrically shrinking regularization. The result is an algorithm that solves $\widetilde{O}(\mathrm{poly}(p)n^{1/3})$ linear systems. For $p = \omega(1)$, this matches the state-of-the-art $n$ dependence [1] but obtains worse dependence on $p$. Nevertheless, we provide a straightforward alternative approach to prior work and our results leave room for further refinements which we believe may result in stronger guarantees.

## 1.2  Related work

Our developments are rooted in three lines of work, which we now briefly survey.

**Monteiro-Svaiter framework instantiations.** Monteiro and Svaiter [25] propose a new acceleration framework, which they specialize to recover the classic fast gradient method [27] and obtain an optimal accelerated second-order method for convex problems with Lipschitz Hessian. Subsequent work [19] extends this to functions with $p$th-order Lipschitz derivatives and a $p$th-order oracle. Generalizing further, Bubeck et al. [9] implement the MS oracle via a "$\Phi$ prox" oracle that given query $x$ returns roughly $\arg\min_{x'}\{f(x) + \Phi(\|x' - x\|)\}$, for continuously differentiable $\Phi$, and prove an error bound scaling with the iterate number $k$ as $\phi(R/k^{3/2})R^2/k^2$, where $\phi(t) = \Phi'(t)/t$. Using $\mathrm{poly}(d)$ parallel queries to a subgradient oracle for non-smooth $f$, they show how to implement the $\Phi$ prox oracle for $\Phi(t) \propto (t/r)^p$ with arbitrarily large $p$, where $r = \epsilon/\sqrt{d}$. Our notion of a ball optimization corresponds to taking $p = \infty$, i.e., letting $\Phi$ be the indicator of $[0, r]$. However, since such $\Phi$ is not continuous, our result does not follow directly from [9]. Thus, our approach clarifies the limiting behavior of MS acceleration of infinitely smooth functions.

**Trust region methods.** The idea of approximately minimizing the objective in a "trust region" around the current iterate plays a central role in nonlinear optimization and machine learning [see, e.g., 15, 23, 28]. Typically, the approximation takes the form of a second-order Taylor expansion, where regularity of the Hessian is key for guaranteeing the approximation quality. Of particular relevance to us is the work of Karimireddy et al. [22], which define a notion of Hessian stability under which a trust-region method converges linearly with only logarithmic dependence on problem conditioning. We observe that this stability condition in fact renders the second-order trust region approximation highly effective, so that a few iterations suffice in order to implement an "ideal" ball optimization oracle, thus enabling accelerated condition-free convergence.

Karimireddy et al. [22] also observe that quasi self-concordance (QSC) is a sufficient condition for Hessian stability, and that the logistic regression objective is QSC. We use this observation for our applications, and prove that the softmax objective is also QSC. Marteau-Ferey et al. [24] directly leverage the QSC property using Newton method variants. For QSC functions with parameter $M$, they show complexity guarantees scaling linearly in $MR$. Under the same assumptions, we obtain the improved scaling $(MR)^{2/3}$. Both guarantees depend only weakly (polylogarithmically) on the standard problem condition number.

**Efficient $\ell_p$ regression algorithms.** There has been rapid recent progress in linearly convergent algorithms for minimizing the $p$-norm of the regression residual $\mathbf{A}x - b$ or alternatively for finding a minimum $p$-norm $x$ satisfying linear constraints $\mathbf{A}x = b$. Bubeck et al. [8] give faster algorithms for all $p \in (1, 2) \cup (2, \infty)$, discovering and overcoming a limitation of classic interior point methods. Their algorithm is based on considering a smooth interpolation between a quadratic and the original objective. Bullins [10] applies accelerated tensor methods to develop a gradient descent method for the case of $p = 4$ with linear-system solution complexity scaling as $n^{1/5}$ (for $\mathbf{A} \in \mathbb{R}^{n \times d}$). Adil et al. [2] give an iterative refinement method for general $p \in (1, \infty)$ with complexity proportional to $n^{|p-2|/(2p+|p-2|)} \leq n^{1/3}$, matching [10] for $p = 4$ and improving on [8]. Adil and Sachdeva [1] provide an alternative method with complexity scaling as $p \cdot n^{1/3}$ scaling, improving on the $O(p^{O(p)})$ dependence in [2].

As mentioned in the previous section, a number of recent works [11, 18, 13] obtain $\epsilon$-accurate solutions for $p = \infty$ with complexity scaling polynomially in $\epsilon^{-1}$. Bullins and Peng [11] leverage accelerated tensor methods and fourth-order smoothness, Ene and Vladu [18] carefully analyze re-weighted least squares, and Carmon et al. [13] develop a first-order stochastic variance reduction technique for matrix saddle-point problems. We believe that our approach brings us closer to a unified perspective on high-order smoothness and acceleration for regression problems.

## 1.3 Paper organization

In Section 2, we implement the MS oracle using a ball optimization oracle and prove its $\widetilde{O}((R/r)^{2/3})$ convergence guarantee. In Section 3, we show how to use Hessian stability to efficiently implement a ball optimization oracle, and also show that quasi-self-concordance implies Hessian stability. In Section 4 we apply our developments to the aforementioned regression tasks. Finally, in Section 5 we give a lower bound implying our oracle complexity is optimal (up to logarithmic terms).

**Notation.** Let $\mathbf{M}$ be a positive semidefinite matrix, and let $\mathbf{M}^{\dagger}$ be its pseudoinverse. We perform our analysis in the Euclidean seminorm $\|x\|_{\mathbf{M}} \stackrel{\text{def}}{=} \sqrt{x^{\top}\mathbf{M}x}$; we will choose a specific $\mathbf{M}$ when discussing applications. We denote the $\|\cdot\|_{\mathbf{M}}$ ball of radius $r$ around $\bar{x}$ by $\mathcal{B}_r(\bar{x}) \stackrel{\text{def}}{=} \{x \in \mathbb{R}^d \mid \|x - \bar{x}\|_{\mathbf{M}} \leq r\}$. We recall standard definitions of smoothness and strong-convexity in a quadratic norm: differentiable $f : \mathbb{R}^d \to \mathbb{R}$ is $L$-smooth in $\|\cdot\|_{\mathbf{M}}$ if its gradient is $L$-Lipschitz in $\|\cdot\|_{\mathbf{M}}$, and twice-differentiable $f$ is $L$-smooth and $\mu$-strongly convex in $\|\cdot\|_{\mathbf{M}}$ if $\mu\mathbf{M} \preceq \nabla^2 f(x) \preceq L\mathbf{M}$ for all $x \in \mathbb{R}^d$.

# 2 Monteiro-Svaiter Acceleration with a Ball Optimization Oracle

In this section, we give an accelerated algorithm for optimization with the following oracle.

**Definition 1** (Ball optimization oracle). We call $\mathcal{O}_{\text{ball}}$ a $(\delta, r)$-*ball optimization oracle* for $f : \mathbb{R}^d \to \mathbb{R}$ if for any $\bar{x} \in \mathbb{R}^d$, it outputs $y = \mathcal{O}_{\text{ball}}(\bar{x}) \in \mathcal{B}_r(\bar{x})$ such that $\|y - x_{\bar{x},r}\|_{\mathbf{M}} \leq \delta$ for some $x_{\bar{x},r} \in \arg\min_{x \in \mathcal{B}_r(\bar{x})} f(x)$.

We use the Monteiro and Svaiter acceleration framework [25, 19, 9], relying on the following oracle.

**Definition 2** (MS oracle). We call $\mathcal{O}_{\text{MS}}$ a $\sigma$-MS oracle for differentiable $f : \mathbb{R}^d \to \mathbb{R}$ if given inputs $(A, x, v) \in \mathbb{R}_{\geq 0} \times \mathbb{R}^d \times \mathbb{R}^d$, $\mathcal{O}_{\text{MS}}$ outputs $(\lambda, a_\lambda, y_{t_\lambda}, z) \in \mathbb{R}_{\geq 0} \times \mathbb{R}_{\geq 0} \times \mathbb{R}^d \times \mathbb{R}^d$ such that

$$a_\lambda = \frac{\lambda + \sqrt{\lambda^2 + 4\lambda A}}{2}, \ t_\lambda = \frac{A}{A + a_\lambda}, \ y_{t_\lambda} = t_\lambda \cdot x + (1 - t_\lambda) \cdot v,$$

and we have the guarantee

$$\left\|z - (y_{t_\lambda} - \lambda\mathbf{M}^{\dagger}\nabla f(z))\right\|_{\mathbf{M}} \leq \sigma \left\|z - y_{t_\lambda}\right\|_{\mathbf{M}}. \tag{3}$$

We now state the acceleration framework and the main bound we use to analyze its convergence.

---

**Algorithm 1** Monteiro-Svaiter acceleration

---

1: **Input:** Strictly convex and differentiable function $f : \mathbb{R}^d \to \mathbb{R}$. Symmetric $\mathbf{M} \succeq 0$ with $\nabla f(x) \in \text{Im}(\mathbf{M})$ for all $x \in \mathbb{R}^d$. Initialization $A_0 \geq 0$ and $x_0 = v_0 \in \mathbb{R}^d$. Monteiro-Svaiter oracle $\mathcal{O}_{\text{MS}}$ with parameter $\sigma \in [0, 1)$.
2: **for** $k = 0, 1, 2, \dots$ **do**
3: $\quad (\lambda_{k+1}, a_{k+1}, y_k, x_{k+1}) \leftarrow \mathcal{O}_{\text{MS}}(A_k, x_k, v_k)$
4: $\quad v_{k+1} \leftarrow v_k - a_{k+1}\mathbf{M}^{\dagger}\nabla f(x_{k+1}), \qquad A_{k+1} \leftarrow A_k + a_{k+1}$.
5: **end for**

---

**Proposition 3.** *Let differentiable $f$ be strictly convex, $\|x_0 - x^*\|_{\mathbf{M}} \leq R$ and $f(x_0) - f(x^*) \leq \epsilon_0$. Set $A_0 = R^2/(2\epsilon_0)$ and suppose that for some $r > 0$ the iterates of Algorithm 1 satisfy $\|x_{k+1} - y_k\|_{\mathbf{M}} \geq r$ for all $k \geq 0$. Then, the iterates also satisfy $f(x_k) - f(x^*) \leq 2\epsilon_0 \exp(-(\frac{r(1-\sigma)}{R})^{2/3}(k-1))$.*

Proposition 3 is one of our main technical results, obtained via applying a reverse Hölder's inequality on a variant of the performance guarantees of [25]; we defer the proof to Appendix B. Clearly, Proposition 3 implies that the progress of Algorithm 1 is related to the amount of movement of the iterates, i.e., the quantities $\{\|x_{k+1} - y_k\|_{\mathbf{M}}\}$. We now show that by using a ball optimization oracle of radius $r$, we are able to guarantee movement by roughly $r$, which implies rapid convergence. We rely on the following characterization, whose proof we defer to Appendix C.

**Lemma 4.** *Let* $f : \mathbb{R}^d \to \mathbb{R}$ *be continuously differentiable and strictly convex. For all* $y \in \mathbb{R}^d$, $z = \arg\min_{z' \in \mathcal{B}_r(y)} f(z')$ *either globally minimizes* $f$, *or* $\|z - y\|_{\mathbf{M}} = r$ *and* $\nabla f(z) = -\frac{\|\nabla f(z)\|_{\mathbf{M}^\dagger}}{r} \mathbf{M}(z - y)$.

Lemma 4 implies that a $(0, r)$ ball optimization oracle either globally minimizes $f$, or yields $z$ with

$$\|z - y\|_{\mathbf{M}} = r \text{ and } \|z - (y - \lambda \mathbf{M}^\dagger \nabla f(z))\|_{\mathbf{M}} = 0, \quad \text{for } \lambda = \frac{r}{\|\nabla f(z)\|_{\mathbf{M}^\dagger}}. \quad (4)$$

This is precisely the type of bound compatible with both Proposition 3 and requirement (3) of $\mathcal{O}_{\mathrm{MS}}$. The remaining difficulty lies in that $\lambda$ also defines the point $y = y_{t_\lambda}$. Therefore, to implement an MS oracle using a ball optimization oracle we perform binary search over $\lambda$, with the goal of solving

$$g(\lambda) \stackrel{\text{def}}{=} \lambda \|\nabla f(z_{t_\lambda})\|_{\mathbf{M}^\dagger} = r, \text{ where } z_{t_\lambda} \stackrel{\text{def}}{=} \min_{z \in \mathcal{B}_r(y_{t_\lambda})} f(z), \text{ and } t_\lambda, y_{t_\lambda} \text{ as in Definition 2}.$$

Algorithm 2 describes our binary search implementation. The algorithm takes the MS oracle input $(A, x, v)$ as well $D$ bounding the distance of $x$ and $v$ from the optimum, and desired global solution accuracy $\epsilon$, outputting either a (global) $\epsilon$-approximate minimizer or $(\lambda, a_\lambda, y_{t_\lambda}, \tilde{z}_{t_\lambda})$ satisfying both (3) (with $\sigma = \frac{1}{2}$) and a lower bound on $\|\tilde{z}_{t_\lambda} - y_{t_\lambda}\|_2$. To bound our procedure's complexity we leverage $L$-smoothness of $f$ (i.e. $L$-Lipschitz continuity of $\nabla f$), yielding a bound on the Lipschitz constant of $g(\lambda)$ defined above. Our analysis is somewhat intricate as it must account for inexactness in the ball optimization oracle. It obtains the following performance guarantee, whose proof is in Appendix C.

**Proposition 5** (Guarantees of Algorithm 2). *Let* $L, D, \delta, r > 0$ *and* $\mathcal{O}_{\mathrm{ball}}$ *satisfy the requirements in Lines 1–3 of Algorithm 2, and* $\epsilon < 2LD^2$. *Then, Algorithm 2 either returns* $\tilde{z}_{t_\lambda}$ *with* $f(\tilde{z}_{t_\lambda}) - f(x^*) < \epsilon$, *or implements a* $\frac{1}{2}$-*MS oracle with the additional guarantee* $\|\tilde{z}_{t_\lambda} - y_{t_\lambda}\|_{\mathbf{M}} \geq \frac{11r}{12}$. *Moreover, the number of calls to* $\mathcal{O}_{\mathrm{ball}}$ *is bounded by* $O(\log(\frac{LD^2}{\epsilon}))$.

---

**Algorithm 2** Monteiro-Svaiter oracle implementation

1: **Input:** Function $f : \mathbb{R}^d \to \mathbb{R}$ that is strictly convex, $L$-smooth in $\|\cdot\|_{\mathbf{M}}$. $A \in \mathbb{R}_{\geq 0}$ and $x, v \in \mathbb{R}^d$ satisfying $\|x - x^*\|_{\mathbf{M}}$ and $\|v - x^*\|_{\mathbf{M}} \leq D$ where $x^* = \arg\min_x f(x)$. A $(\delta, r)$-ball optimization oracle $\mathcal{O}_{\mathrm{ball}}$, where $\delta = \frac{r}{12(1+Lu)}$ and $u = \frac{2(D+r)r}{\epsilon}$.
2: Set $\lambda \leftarrow u$ and $\ell \leftarrow \frac{r}{2LD}$, let $\tilde{z}_{t_\lambda} \leftarrow \mathcal{O}_{\mathrm{ball}}(y_{t_\lambda})$
3: **if** $u \|\nabla f(\tilde{z}_{t_\lambda})\|_{\mathbf{M}^\dagger} \leq r + uL\delta$ **then**
4:     **return** $(\lambda, a_\lambda, y_{t_\lambda}, \tilde{z}_{t_\lambda})$
5: **else**
6:     **while** $|\lambda \|\nabla f(\tilde{z}_{t_\lambda})\|_{\mathbf{M}^\dagger} - r| > \frac{r}{6}$ **do**
7:         $\lambda \leftarrow \frac{\ell + u}{2}$, $\tilde{z}_{t_\lambda} \leftarrow \mathcal{O}_{\mathrm{ball}}(y_{t_\lambda})$
8:         **if** $\lambda \|\nabla f(\tilde{z}_{t_\lambda})\|_{\mathbf{M}^\dagger} \geq r$ **then** $u \leftarrow \lambda$, **else** $\ell \leftarrow \lambda$
9:     **end while**
10:     **return** $(\lambda, a_\lambda, y_{t_\lambda}, \tilde{z}_{t_\lambda})$
11: **end if**

---

Finally, we state our main acceleration result, whose proof we defer to Appendix C.

**Theorem 6** (Acceleration with a ball optimization oracle). *Let* $\mathcal{O}_{\mathrm{ball}}$ *be an* $\left(\frac{r}{12+126LRr/\epsilon}, r\right)$-*ball optimization oracle for strictly convex and* $L$-*smooth* $f : \mathbb{R}^d \to \mathbb{R}$ *with minimizer* $x^*$, *and initial point* $x_0$ *satisfying* $\|x_0 - x^*\|_{\mathbf{M}} \leq R$ *and* $f(x_0) - f(x^*) \leq \epsilon_0$. *Then, Algorithm 1 using Algorithm 2 as a Monteiro-Svaiter oracle with* $D = \sqrt{18}R$ *produces an iterate* $x_k$ *with* $f(x_k) - f(x^*) \leq \epsilon$, *in* $O\left((R/r)^{2/3} \log(\epsilon_0/\epsilon) \log(LR^2/\epsilon)\right)$ *calls to* $\mathcal{O}_{\mathrm{ball}}$.

## 3  Ball Optimization Oracle for Hessian Stable Functions

In this section we leverage standard techniques for solving the trust-region subproblem [15] in order to implement a ball optimization oracle. The key structure enabling efficient implementation is the the following notion of Hessian stability, a slightly stronger version of the condition in Karimireddy et al. [22].[2]

**Definition 7** (Hessian stability). Twice-differentiable $f : \mathbb{R}^d \to \mathbb{R}$ is $(r, c)$-*Hessian stable* for $r, c \geq 0$ with respect to $\|\cdot\|$ if $\forall x, y \in \mathbb{R}^d$ with $\|x - y\| \leq r$ we have $c^{-1} \nabla^2 f(y) \preceq \nabla^2 f(x) \preceq c \nabla^2 f(y)$.

We give a method implementing a $(\delta, r)$-ball oracle (cf. Definition 1) for $(r, c)$-stable functions in $\|\cdot\|_{\mathbf{M}}$, requiring $\widetilde{O}(c)$ linear system solutions. The method reduces the oracle to solving $\widetilde{O}(c)$ trust-region subproblems of the form $\min_{x \in \mathcal{B}_r(\bar{x})} Q(x) \overset{\text{def}}{=} -g^\top x + \frac{1}{2} x^\top \mathbf{H} x$, and we show each requires $\widetilde{O}(1)$ linear system solves in $\mathbf{H} + \lambda \mathbf{M}$ for $\lambda \geq 0$. In terms of total linear system solves, our method has a (mild) polylogarithmic dependence on the *condition number* of $f$ in $\|\cdot\|_{\mathbf{M}}$. The main result of this section is Theorem 8, which guarantees correctness and complexity our ball optimization oracle implementation; proofs are deferred to Appendices D.1 and D.2.

**Theorem 8.** *Let $f$ be $L$-smooth, $\mu$-strongly convex, and $(r, c)$-Hessian stable in the seminorm $\|\cdot\|_{\mathbf{M}}$. Then, Algorithm 7 (in Appendix D.2) implements a $(\delta, r)$-ball optimization oracle for query point $\bar{x}$ with $\|\bar{x} - x^*\|_{\mathbf{M}} \leq D$ for $x^*$ the minimizer of $f$, and requires*

$$O\left( c \log^2 \left( \frac{\kappa(D + r)c}{\delta} \right) \right)$$

*linear system solves in matrices of the form $\mathbf{H} + \lambda \mathbf{M}$ for nonnegative $\lambda$, where $\kappa = L/\mu$.*

**Remark 9** (First-order implementation). *The linear system solves required by Theorem 8 can be carried out via Gaussian elimination, fast matrix multiplication, or a number of more scalable algorithms, including first-order methods [e.g., 5]. In Section 4.3, we show that using first-order methods that exploit the particular problem structure allows us to achieve state-of-the-art runtimes for $\ell_\infty$ regression in certain regimes.*

We state a sufficient condition for Hessian stability below. We use this result in Section 4 to establish Hessian stability in several structured problems, and defer its proof to Appendix E for completeness.

**Definition 10** (Quasi-self-concordance). We say that thrice-differentiable $f : \mathbb{R}^d \to \mathbb{R}$ is $M$-*quasi-self-concordant* (QSC) with respect to some norm $\|\cdot\|$, for $M \geq 0$, if for all $u, h, x \in \mathbb{R}^d$,

$$\left| \nabla^3 f(x)[u, u, h] \right| \leq M \|h\| \|u\|^2_{\nabla^2 f(x)},$$

i.e., restricting the third-derivative tensor of $f$ to any direction is bounded by a multiple of its Hessian.

**Lemma 11.** *If thrice-differentiable $f : \mathbb{R}^d \to \mathbb{R}$ is $M$-quasi-self-concordant with respect to norm $\|\cdot\|$, then it is $(r, \exp(Mr))$-Hessian stable with respect to $\|\cdot\|$.*

## 4 Applications

Algorithm 3 puts together the ingredients from previous sections to give a complete second-order method for minimizing QSC functions. We now apply it to functions of the form $f(x) = g(\mathbf{A}x)$ for matrix $\mathbf{A} \in \mathbb{R}^{n \times d}$ and $g : \mathbb{R}^n \to \mathbb{R}$. The logistic loss, softmax approximation of $\ell_\infty$ regression, and variations of $\ell_p$ regression objectives all have this form. The following complexity guarantee for Algorithm 3 follows directly from our previous developments and we defer a proof to Appendix F.

---

**Algorithm 3** Monteiro-Svaiter accelerated BAll COnstrained Newton's method (`MS-BACON`)

---

1: **Input:** Function $f : \mathbb{R}^d \to \mathbb{R}$, desired accuracy $\epsilon$, initial point $x_0$, initial suboptimality $\epsilon_0$.
2: **Input:** Domain bound $R$, quasi-self-concordance $M$, smoothness $L$, norm $\|\cdot\|_{\mathbf{M}}$.
3: Define $\tilde{f}(x) = f(x) + \frac{\epsilon}{55R^2} \|x - x_0\|^2_{\mathbf{M}}$
4: Using Algorithm 7, implement $\mathcal{O}_{\text{ball}}$, a $(\delta, \frac{1}{M})$-ball optimization oracle for $\tilde{f}$, where $\delta = \Theta(\frac{\epsilon}{LR})$
5: Using Algorithm 2 and $\mathcal{O}_{\text{ball}}$, implement $\mathcal{O}_{\text{MS}}$, a $\frac{1}{2}$-MS oracle for $\tilde{f}$
6: Using $O((RM)^{2/3} \log \frac{\epsilon_0}{\epsilon})$ iterations of Algorithm 1 with $\mathcal{O}_{\text{MS}}$ and initial point $x_0$ compute $x_{\text{out}}$, an $\epsilon/2$-accurate minimizer of $\tilde{f}$
7: **return** $x_{\text{out}}$

---

**Corollary 12.** *Let $f(x) = g(\mathbf{A}x)$, for $g : \mathbb{R}^n \to \mathbb{R}$ that is $L$-smooth, $M$-QSC in the $\ell_2$ norm, and $\mathbf{A} \in \mathbb{R}^{n \times d}$. Let $x^*$ be a minimizer of $f$, and suppose that $\|x_0 - x^*\|_{\mathbf{M}} \leq R$ and $f(x_0) - f(x^*) \leq \epsilon_0$*

*for some $x_0 \in \mathbb{R}^d$, where $\mathbf{M} \stackrel{\text{def}}{=} \mathbf{A}^\top \mathbf{A}$. Then, Algorithm 3 yields an $\epsilon$-approximate minimizer to $f$ in*

$$O\left((RM)^{2/3} \log\left(\frac{\epsilon_0}{\epsilon}\right) \log^3\left(\frac{LR^2}{\epsilon}(1 + RM)\right)\right)$$

*linear system solves in matrices of the form $\mathbf{A}^\top \left(\nabla^2 g(\mathbf{A}x) + \lambda \mathbf{I}\right) \mathbf{A}$ for $\lambda > 0$ and $x \in \mathbb{R}^d$.*

Both the (unaccelerated) Newton method-based algorithm in Marteau-Ferey et al. [24] and our method depend polylogarithmically on the (regularized) problem's condition number. The method proposed in Marteau-Ferey et al. [24] has a complexity of $\widetilde{O}(MR)$ for solving $M$-QSC functions with domain size $R$, while our method gives an accelerated dependence of $\widetilde{O}((MR)^{2/3})$. We defer proofs of claims in the following subsections to Appendix F.

## 4.1  Logistic regression

Consider *logistic regression* in matrix $\mathbf{A} \in \mathbb{R}^{n \times d}$ with $n$ data points of dimension $d$, and corresponding labels $b \in \{-1, 1\}^n$. The objective is

$$f(x) = \sum_{i \in [n]} \log(1 + \exp(-b_i\langle a_i, x\rangle)) = g(\mathbf{A}x), \tag{5}$$

where $g(y) = \sum_{i \in [n]} \log(1 + \exp(-b_i y_i))$. It is known [6] that $g$ is 1-QSC and 1-smooth in $\ell_2$, with a diagonal Hessian. Thus, we have the following convergence guarantee from Corollary 12.

**Corollary 13.** *For the logistic regression objective* (5), *given $x_0$ with initial function error $\epsilon_0$ with distance $R$ from a minimizer in $\|\cdot\|_{\mathbf{A}^\top \mathbf{A}}$, Algorithm 3 obtains an $\epsilon$-approximate minimizer using $O\left(R^{2/3} \log\left(\epsilon_0/\epsilon\right) \log^3\left(R^2(1+R)/\epsilon\right)\right)$ linear system solves in matrices $\mathbf{A}^\top \mathbf{D} \mathbf{A}$ for diagonal $\mathbf{D}$.*

Compared to Karimireddy et al. [22], which gives a trust-region Newton method using $\widetilde{O}(R)$ linear system solves, we obtain an improved dependence on the domain size $R$.

## 4.2  $\ell_\infty$ regression

Consider $\ell_\infty$ *regression* in matrix $\mathbf{A} \in \mathbb{R}^{n \times d}$ and vector $b \in \mathbb{R}^n$, which asks to minimize

$$f(x) = \|\mathbf{A}x - b\|_\infty = g(\mathbf{A}x), \tag{6}$$

where $g(y) = \|y - b\|_\infty$. Without loss of generality (by concatenating $\mathbf{A}$, $b$ with $-\mathbf{A}$, $-b$), we may replace the $\|\cdot\|_\infty$ in the objective with a maximum. It is well-known that $g(y)$ is approximated within additive $\epsilon/2$ by $\text{lse}_t(y - b)$ for $t = \epsilon/(2\log n)$ (see Lemma 42 for a proof), where we set

$$\text{lse}(x) \stackrel{\text{def}}{=} \log\left(\sum_{i \in [n]} \exp(x_i)\right), \quad \text{lse}_t(x) \stackrel{\text{def}}{=} t\,\text{lse}(x/t).$$

Our improvement stems from the fact that $\text{lse}_t$ is QSC, which appears to be a new observation. The proof carefully manipulates the third-derivative tensor of $\text{lse}_t$ and is deferred to Appendix F.

**Lemma 14.** $\text{lse}_t$ *is $1/t$-smooth and $2/t$-QSC in $\ell_\infty$.*

Lemma 14 immediately implies that $\text{lse}_t$ is $n/t$-smooth and $2/t$-QSC in $\ell_2$. We thus obtain the following by applying Corollary 12 to the $\text{lse}_{\epsilon/(2\log n)}$ objective, and solving to $\epsilon/2$ additive accuracy.

**Corollary 15.** *Given $x_0$ with initial function error $\epsilon_0$ with distance $R$ from a minimizer in $\|\cdot\|_{\mathbf{A}^\top \mathbf{A}}$, Algorithm 3 obtains an $\epsilon$-approximate minimizer using $O\left((R\log n/\epsilon)^{2/3} \log\left(\epsilon_0/\epsilon\right) \log^3\left(nR/\epsilon\right)\right)$ linear system solves in matrices $\hat{\mathbf{A}}^\top \mathbf{D} \hat{\mathbf{A}}$, where $\mathbf{D}$ is a positive definite diagonal matrix, and $\hat{\mathbf{A}}$ is the vertical concatenation of $\mathbf{A}$ and $-\mathbf{A}$.*

The reduction from solving linear systems of the form described in Corollary 12 to linear systems of the form in Corollary 15 (which is not immediate, since the Hessian of softmax is not diagonal) is given in Appendix F.

Compared to Bullins and Peng [11], which find an $\epsilon$-approximate solution to (6) in $\widetilde{O}((R/\epsilon)^{4/5})$ linear system solves using high-order acceleration, we obtain an improved dependence on $R/\epsilon$. Ene

and Vladu [18] consider the equivalent problem $\text{minimize}_{y:\mathbf{A}^\top y=c}\|y\|_\infty$ (see Appendix F.2.1 for explanation of this equivalence). They show how to solve this problem to $\delta$ multiplicatie error in $\widetilde{O}(n^{1/3}\delta^{-2/3})$ linear system solutions in $\mathbf{A}^\top \mathbf{D}\mathbf{A}$ for positive diagonal $\mathbf{D}$. Translated into our setting, this implies a complexity of $\widetilde{O}(n^{1/3}\|\mathbf{A}x^*\|_\infty^{2/3}\epsilon^{-2/3})$ linear system solves in $\mathbf{A}^\top \mathbf{D}\mathbf{A}$, which is never better than our guarantee since $\|v\|_2 \le \sqrt{n}\|v\|_\infty$ for all $v \in \mathbb{R}^n$. Conversely, our result maps to the setting of Ene and Vladu [18] to provide a complexity guarantee of $\widetilde{O}(\|x^*\|_2^{2/3}\epsilon^{-2/3})$ appropriate linear system solves to attain $\epsilon$ additive error.

Finally, we note that our unconstrained regression solver also solves constrained regression problems which are sometimes considered in the literature, through a reduction.

### 4.3 First-order methods and improved norm dependence

For both logistic regression and $\ell_\infty$ regression, we can alternatively work in the standard $\ell_2$ norm, and obtain a different QSC parameter depending on $\max_i\|a_i\|_2$; we defer all proofs to Appendix F.3.

**Lemma 16.** *The logistic objective $f(x) = g(\mathbf{A}x)$ in (5) is $\max_{i\in[n]}\|a_i\|_2$-QSC in the $\ell_2$ norm.*

**Lemma 17.** *The log-sum-exp function $f(x) = \text{lse}_t(\mathbf{A}x)$ is $\frac{2}{t}\max_{i\in[n]}\|a_i\|_2$-QSC in the $\ell_2$ norm.*

With these alternative QSC bounds, we turn our attention to the cost of implementing a ball oracle. In the previous sections we accomplish this by using a generic positive semidefinite linear system solver; we now demonstrate how first-order methods can give improved runtimes in large-scale settings. We focus on $\ell_\infty$ regression here, as the case of logistic regression is similar. Defining $R = \|x_0 - x^*\|_2$, we seek an $\epsilon/4$-approximate minimizer to a smooth, strongly-convex approximation of the $\ell_\infty$-norm: we pick

$$h(x) = \text{lse}_t(\mathbf{A}x) + \frac{\epsilon}{4R^2}\|x - x_0\|_2^2, \text{ where } t = \frac{\epsilon}{2\log n}.$$

By applying variance-reduced stochastic gradient methods to solve linear systems in $\nabla^2 h(x)$ and combining with our framework, we obtain the following complexity bound in terms of runtime (as opposed to linear system solves).

**Corollary 18.** *With initial function error $\epsilon_0$ and $R = \|x_0 - x^*\|_2$, Algorithm 3 using the first-order linear system solver of Agarwal et al. [5] returns an $\epsilon$-approximate minimizer within total runtime $\widetilde{O}\left(\left(\max_{i\in[n]}\|a_i\|_2\frac{R}{\epsilon}\right)^{2/3}\left(nd + d^{1.5}\max_{i\in[n]}\|a_i\|_2\frac{R}{\epsilon}\right)\right).$*

Let $L = \max_{i\in[n]}\|a_i\|_2$. In the regime $d \le (\frac{LR}{\epsilon})^{2/3} \le \frac{n}{d}$ and when $\mathbf{A}$ is dense, we obtain a speed-up compared to the state-of-the-art runtime $\widetilde{O}(nd + \sqrt{nd(n+d)}\frac{LR^2}{\epsilon})$ of Carmon et al. [13].

### 4.4 $\ell_p$ regression

Consider $\ell_p$ regression in matrix $\mathbf{A} \in \mathbb{R}^{n\times d}$ and vector $b \in \mathbb{R}^n$, which asks to minimize

$$f(x) = \|\mathbf{A}x - b\|_p^p = g(\mathbf{A}x) \text{ with optimizer } x^*. \tag{7}$$

for some fixed $p > 3$,[3] where $g(x) = \sum_i|x_i - b_i|^p$. While this objective is not QSC, our method iteratively considers a regularized QSC objective to halve the error, as summarized in Algorithm 8.

---

**Algorithm 4** High accuracy $\ell_p$ regression

---

1: **Input:** $\mathbf{A} \in \mathbb{R}^{n\times d}, b \in \mathbb{R}^n$, multiplicative error tolerance $\delta \ge 0$.
2: Set $x_0 = \mathbf{A}^\dagger b$ and $\epsilon_0 = f(x_0) = \|\mathbf{A}x_0 - b\|_p^p$.
3: **for** $k \le \log_2(n/\delta^{1/p})$ **do**
4: $\quad\epsilon_k \leftarrow 2^{-p}\epsilon_{k-1}$
5: $\quad x_k \leftarrow$ output of Algorithm 3 applied on $f(x) = \|\mathbf{A}x - b\|_p^p$ with initialization $x_{k-1}$, desired
$\quad\quad$ accuracy $\epsilon_k$ and parameters $R = O(n^{(p-2)/2p}\epsilon_k^{1/p})$ and $M = O(p\sqrt{n}/R)$ (see Lemma 52)
6: **end for**

---

Below we state the guarantee of Algorithm 8, and defer its proof to Appendix F.4.

**Corollary 19.** *Algorithm 8 computes $x \in \mathbb{R}^d$ with $\|\mathbf{A}x - b\|_p^p \leq (1 + \delta)\|\mathbf{A}x^* - b\|_p^p$ using $O(p^{14/3}n^{1/3}\log^4(n/\delta))$ linear system solves in $\mathbf{A}^\top \mathbf{D}\mathbf{A}$ for diagonal matrix $\mathbf{D} \succeq 0$.*

Compared to Adil and Sachdeva [1], Adil et al. [2], which minimize $f$ to $1+\delta$ multiplicative accuracy by solving $\widetilde{O}\left(\min\left(pn^{1/3}, p^{O(p)}n^{\frac{p-2}{3p-2}}\right)\log(1/\delta)\right)$ linear systems, our guarantee is slightly weaker in its $p$ dependence. Nonetheless, we believe our alternative, simple approach sheds further light on the complexity of this problem, and that there is room for additional improvement.

## 5    Lower Bound

We establish a lower bound showing that all algorithms for minimizing a function via repeated calls to a $(0, r)$ ball optimization oracle, which we call *r-BOO algorithms*, require $\Omega((R/r)^{2/3})$ queries in the worst case. Formally, the following lower bound matches Theorem 6 up to logarithmic factors.

**Theorem 20.** *Let $\frac{r}{R}, \delta \in (0, 1)$ and $d = \left\lceil 60(\frac{R}{r})^2 \log \frac{R}{\delta \cdot r} \right\rceil$. There exists a distribution $P$ over convex and 1-Lipschitz functions from $\mathcal{B}_R(0) \to \mathbb{R}$ such that the following holds for any $r$-BOO algorithm: with probability at least $1 - \delta$ over the draw of $f \sim P$ and the algorithm's coin flips (i.e. randomness used by the algorithm), its first $\left\lceil \frac{1}{10}(\frac{R}{r})^{2/3} \right\rceil$ queries are at least $R^{2/3}r^{1/3}$ suboptimal for $f$.*

We prove Theorem 20 in Appendix G as a corollary of a stronger results stating the same lower bound for *r-local oracle* algorithms, that for each query $x$ receive the function $f$ restricted to $\mathcal{B}_r(x)$. This information clearly suffices to compute the ball optimization oracle output as well as $f(x)$ and $\nabla f(x)$, implying that Algorithm 2 also operates within our oracle framework. Our proof is essentially a careful reading of the classical information-based complexity lower-bound for convex optimization [26, 20], where we strengthen the notion of local oracles—which return $f$ restricted to a neighborhood of the query—by quantifying the size of the neighborhood.

Using arguments from [20, 17] we may further strengthen the lower bound to hold for instances which are smooth, strongly-convex and have unbounded domain, precisely matching the assumptions of Theorem 6 (see Appendix G). However, the *implementations* of the ball optimization oracle we consider in Section 3 require a Hessian stability assumption (Definition 7), and it is unclear if we can make the hard instances underlying Theorem 20 Hessian-stable. Nevertheless, our lower bound precludes further progress via the ball optimization oracle abstraction, up to logarithmic factors.

## Broader Impact

This work does not present any foreseeable societal consequence.

## Acknowledgments

We thank Sébastien Bubeck for helpful conversations.

## Sources of funding

Researchers on this project were supported by NSF CAREER Award CCF-1844855 and CCF-1749609, NSF Grant CCF-1955039, CCF-1740551, DMS-1839116 and DMS-2023166, two Sloan Research Fellowships and Packard Fellowships, and two Stanford Graduate Fellowships. Additional support was provided by PayPal and Microsoft, including two Microsoft Research Faculty Fellowships and a PayPal research gift.

## Competing interests

The authors declare no competing interests.

## Footnotes

[2] A variant of the algorithm we develop also works under the weaker stability condition. We state the stronger condition as it is simpler, and holds for all our applications.

[3]We assume $p > 3$ for ease of presentation; for $p \le 4$ our runtime is superseded by, e.g., the algorithm of [3].

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
