[Supplementary Material]

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

[4]We note that $\log(n/\delta)$ iterations of our algorithm yield the stronger multiplicative accuracy guarantee of $\delta^p$, without an additional dependence on $p$.

[5]The function $\gamma_p(|y|, \Delta)$ in their setting is at least $\|y\|_p^p$.

[6]Applying Yao's minimax principle [31] we implicitly condition our proof on the random coin tosses of $\mathcal{A}$, which is tantamount to assuming without loss of generality that $\mathcal{A}$ is deterministic.

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

# Supplementary material

## A  Unaccelerated optimization with a ball optimization oracle

Here, we state and analyze the unaccelerated algorithm for optimization of convex function $f$ with access to a ball optimization oracle. For simplicity of exposition, we assume that the oracle $\mathcal{O}_{\mathrm{ball}}$ is a $(0, r)$-oracle, i.e. is exact, and we perform our analysis in the $\ell_2$ norm; for a general Euclidean seminorm, a change of basis suffices to give the same guarantees.

---

**Algorithm 5** Iterating ball optimization

---

1: **Input:** Function $f : \mathbb{R}^d \to \mathbb{R}$ and initial point $x_0 \in \mathbb{R}^d$.
2: **for** $k = 1, 2, ...$ **do**
3:     $x_k \leftarrow \mathcal{O}_{\mathrm{ball}}(x_{k-1})$
4: **end for**

---

We first note that the distance $\|x_k - x^*\|_2$ is decreasing in $k$.

**Lemma 21.** *For all $x \in \mathbb{R}^d$, $\|\mathcal{O}_{\mathrm{ball}}(x) - x^*\|_2 \le \|x - x^*\|_2$.*

*Proof.* The claim is obvious if $\mathcal{O}_{\mathrm{ball}}(x) = x^*$, so we assume this is not the case. Note that for any $\tilde{x}$ with $\|\tilde{x} - x\|_2 \le r$, if there is any point $\hat{x}$ on the line between $\tilde{x}$ and $x^*$, then by strict convexity $f(\hat{x}) < f(\tilde{x})$. Now, clearly $\mathcal{O}_{\mathrm{ball}}(x)$ lies on the boundary of the ball around $x$, and moreover the angle between the vectors $x - \mathcal{O}_{\mathrm{ball}}(x)$ and $x^* - \mathcal{O}_{\mathrm{ball}}(x)$ must be obtuse, else the line between $\mathcal{O}_{\mathrm{ball}}(x)$ and $x^*$ intersects the ball twice. Thus, by law of cosines $\|\mathcal{O}_{\mathrm{ball}}(x) - x^*\|_2 \le \sqrt{\|x - x^*\|_2^2 - r^2}$, yielding the conclusion. $\square$

**Theorem 22.** *Suppose for some $x_0 \in \mathbb{R}^d$, $f(x_0) - f(x^*) \le \epsilon_0$ and $\|x_0 - x^*\|_2 \le R$, where $x^*$ is the global minimizer of $f$. Algorithm 5 computes an $\epsilon$-approximate minimizer in $O\left(\frac{R}{r} \log \frac{\epsilon_0}{\epsilon}\right)$ calls to $\mathcal{O}_{\mathrm{ball}}$.*

*Proof.* Define $\tilde{x}_k \stackrel{\mathrm{def}}{=} \left(1 - \frac{r}{R}\right) x_{k-1} + \frac{r}{R} x^*$, and note that because $\|x_{k-1} - x^*\|_2 \le R$, $\tilde{x}_k$ is in the ball of radius $r$ around $x_{k-1}$. Thus, convexity yields

$$f(x_k) \le f(\tilde{x}_k) \le \left(1 - \frac{r}{R}\right) f(x_{k-1}) + \frac{r}{R} f(x^*) \Rightarrow f(x_k) - f(x^*) \le \left(1 - \frac{r}{R}\right)(f(x_{k-1}) - f(x^*)).$$

Iteratively applying this inequality yields the conclusion. $\square$

## B  Analysis of Monteiro-Svaiter acceleration

In this section, we prove Proposition 3. We do so by first proving a sequence of lemmas demonstrating properties of Algorithm 1. Throughout, we recall $\nabla f(x) \in \mathrm{Im}(\mathbf{M})$ for all $x$ by assumption. We note that these are variants of existing bounds in the literature [e.g. 25, 9].

**Lemma 23.** *For all $k \ge 0$,*

$$\lambda_{k+1} A_{k+1} = a_{k+1}^2 \text{ and } \sqrt{A_k} \ge \frac{1}{2} \sum_{i \in [k]} \sqrt{\lambda_i}.$$

*Proof.* The first claim is from solving a quadratic in the definition of $a_{k+1}$. The second follows from

$$\sqrt{A_k} \ge \sqrt{A_k} - \sqrt{A_0} = \sum_{i \in [k]} \left(\sqrt{A_i} - \sqrt{A_{i-1}}\right) = \sum_{i \in [k]} \frac{a_i}{\sqrt{A_i} + \sqrt{A_{i-1}}}$$

$$= \sum_{i \in [k]} \frac{\sqrt{\lambda_i A_i}}{\sqrt{A_i} + \sqrt{A_{i-1}}} \ge \frac{1}{2} \sum_{i \in [k]} \sqrt{\lambda_i}$$

where we used that $A_0 \ge 0$ and $\{A_i\}$ are increasing. $\square$

**Lemma 24.** *For all $k \geq 0$, if $\|x_{k+1} - y_k\|_{\mathbf{M}} > 0$, we have for $\sigma \in [0, 1)$,*

$$\|\nabla f(x_{k+1})\|_{\mathbf{M}^\dagger} > 0 \text{ and } \lambda_{k+1} \geq \frac{\|x_{k+1} - y_k\|_{\mathbf{M}}}{\|\nabla f(x_{k+1})\|_{\mathbf{M}^\dagger}} (1 - \sigma) > 0 \,.$$

*Proof.* For the first claim, by (3),

$$\|x_{k+1} - y_k\|_{\mathbf{M}} - \lambda_{k+1}\|\nabla f(x_{k+1})\|_{\mathbf{M}^\dagger} \leq \|x_{k+1} - (y_k - \lambda_{k+1}\mathbf{M}^\dagger \nabla f(x_{k+1}))\|_{\mathbf{M}} \leq \sigma \|x_{k+1} - y_k\|_{\mathbf{M}},$$

since by assumption, for some $\sigma \in [0, 1)$, $\|x_{k+1} - y_k\|_{\mathbf{M}} > 0$, therefore $\|\nabla f(x_{k+1})\|_{\mathbf{M}^\dagger} = 0$ would contradict this assumption.

For the second claim, Cauchy-Schwarz gives

$$\begin{aligned}
\sigma^2 \|x_{k+1} - y_k\|_{\mathbf{M}}^2 &\geq \left\|x_{k+1} - \left(y_k - \lambda_{k+1}\mathbf{M}^\dagger \nabla f(x_{k+1})\right)\right\|_{\mathbf{M}}^2 \\
&\geq \|x_{k+1} - y_k\|_{\mathbf{M}}^2 - 2\lambda_{k+1}\|\nabla f(x_{k+1})\|_{\mathbf{M}^\dagger}\|x_{k+1} - y_k\|_{\mathbf{M}} + \lambda_{k+1}^2\|\nabla f(x_{k+1})\|_{\mathbf{M}^\dagger}^2.
\end{aligned}$$

Solving the quadratic in $\lambda_{k+1}$ implies, for $P \overset{\text{def}}{=} \|\nabla f(x_{k+1})\|_{\mathbf{M}^\dagger}\|x_{k+1} - y_k\|_{\mathbf{M}}$,

$$\lambda_{k+1} \geq \frac{2P - \sqrt{4P^2 - 4(1 - \sigma^2)P^2}}{2\|\nabla f(x_{k+1})\|_{\mathbf{M}^\dagger}^2} = \frac{P(1 - \sigma)}{\|\nabla f(x_{k+1})\|_{\mathbf{M}^\dagger}^2} = \frac{\|x_{k+1} - y_k\|_{\mathbf{M}}}{\|\nabla f(x_{k+1})\|_{\mathbf{M}^\dagger}} (1 - \sigma) \,.$$

$\square$

Next, we provide the following lemma which gives a recursive bound for the potential, $p_k$, which we define as follows:

$$p_k \overset{\text{def}}{=} A_k \epsilon_k + r_k, \text{ where } \epsilon_k \overset{\text{def}}{=} f(x_k) - f(x^*), \; r_k \overset{\text{def}}{=} \frac{1}{2} \|v_k - x^*\|_{\mathbf{M}}^2 \,.$$

We remark that the proof does not use (3) beyond using the property that $a_{k+1} > 0$ (regardless of how they are induced by $\lambda_{k+1}$).

**Lemma 25.** *For all $k \geq 0$,*

$$p_{k+1} \leq p_k + \frac{A_{k+1}^2}{2a_{k+1}^2}\left(\left\|x_{k+1} - \left(y_k - \frac{a_{k+1}^2}{A_{k+1}}\mathbf{M}^\dagger \nabla f(x_{k+1})\right)\right\|_{\mathbf{M}}^2 - \|x_{k+1} - y_k\|_{\mathbf{M}}^2\right).$$

*Proof.* By Lemma 24 we have that $\lambda_{k+1} > 0$, so that $a_{k+1} > 0$. Then,

$$v_k = \frac{1}{a_{k+1}}(A_{k+1}y_k - A_k x_k) = x_{k+1} + \frac{A_{k+1}}{a_{k+1}}(y_k - x_{k+1}) + \frac{A_k}{a_{k+1}}(x_{k+1} - x_k).$$

Consequently, convexity of $f$, i.e., $\langle \nabla f(b), a - b \rangle \leq f(a) - f(b)$ for all $a, b \in \mathbb{R}^n$, yields

$$a_{k+1}\langle \nabla f(x_{k+1}), x^* - v_k \rangle \leq A_k \epsilon_k - A_{k+1}\epsilon_{k+1} + A_{k+1}\langle \nabla f(x_{k+1}), x_{k+1} - y_k \rangle \,.$$

Further, expanding $r_{k+1} = \frac{1}{2}\|v_{k+1} - x_*\|_{\mathbf{M}}^2$, where we recall $v_{k+1} = v_k - a_{k+1}\mathbf{M}^\dagger \nabla f(x_{k+1})$, gives

$$\frac{1}{2}\|v_{k+1} - x^*\|_{\mathbf{M}}^2 = r_k + \frac{a_{k+1}^2}{2}\|\nabla f(x_{k+1})\|_{\mathbf{M}^\dagger}^2 + a_{k+1}\langle \mathbf{M}\mathbf{M}^\dagger \nabla f(x_{k+1}), x^* - v_k \rangle \,.$$

Combining these inequalities, and recalling $\mathbf{M}\mathbf{M}^\dagger \nabla f(x_{k+1}) = \nabla f(x_{k+1})$, then yields that

$$A_{k+1}\epsilon_{k+1} + r_{k+1} \leq A_k \epsilon_k + r_k + \frac{a_{k+1}^2}{2}\|\nabla f(x_{k+1})\|_{\mathbf{M}^\dagger}^2 + A_{k+1}\langle \nabla f(x_{k+1}), x_{k+1} - y_k \rangle \,.$$

The result then follows from $p_k = A_k \epsilon_k + r_k$ and the fact that

$$\frac{A_{k+1}^2}{2a_{k+1}^2}\left\|x_{k+1} - \left(y_k - \frac{a_{k+1}^2}{A_{k+1}}\mathbf{M}^\dagger \nabla f(x_{k+1})\right)\right\|_{\mathbf{M}}^2$$

$$= \frac{A_{k+1}^2}{2a_{k+1}^2}\|x_{k+1} - y_k\|_{\mathbf{M}}^2 + A_{k+1}\langle \nabla f(x_{k+1}), x_{k+1} - y_k \rangle + \frac{a_{k+1}^2}{2}\|\nabla f(x_{k+1})\|_{\mathbf{M}^\dagger}^2 \,.$$

$\square$

Next, we use (3) and the choice of $a_{k+1}$ in the algorithm to improve the bound in Lemma 25.

**Lemma 26.** *For all $k \geq 0$,*

$$p_k + \sum_{i \in [k]} \frac{(1 - \sigma^2) A_i}{2\lambda_i} \|x_{i+1} - y_i\|_{\mathbf{M}}^2 \leq p_0 .$$

*Proof.* Lemma 23 gives that for our choice of parameters, $\lambda_{k+1} A_{k+1} = a_{k+1}^2$ for all $k \geq 0$. Lemma 25 then implies that

$$A_{k+1}\epsilon_{k+1} + r_{k+1} \leq A_k \epsilon_k + r_k + \frac{A_{k+1}}{2\lambda_{k+1}} \left( \left\|x_{k+1} - \left(y_k - \lambda_{k+1}\mathbf{M}^\dagger \nabla f(x_{k+1})\right)\right\|_{\mathbf{M}}^2 - \|x_{k+1} - y_k\|_{\mathbf{M}}^2 \right)$$

$$\leq A_k \epsilon_k + r_k + \frac{(\sigma^2 - 1)A_{k+1}}{2\lambda_{k+1}} \|x_{k+1} - y_k\|_{\mathbf{M}}^2$$

where we used (3) and the claim now follows from inductively applying the resulting bound. $\square$

Below we give a diameter bound on the iterates from the algorithm.

**Lemma 27.** *If $x_0 = v_0$, then for all $k \geq 0$ we have*

$$\|x_k - x^*\|_{\mathbf{M}} \leq \frac{2 - \sigma}{1 - \sigma} \sqrt{2p_0}, \ \|v_k - x^*\|_{\mathbf{M}} \leq \sqrt{2p_0}.$$

*Proof.* Since $p_k = A_k \epsilon_k + r_k$, the second claim follows immediately from Lemma 26 implying that $\frac{1}{2} \|v_k - x^*\|_{\mathbf{M}}^2 = r_k \leq p_0$ for all $k \geq 0$. Further, convexity and the triangle inequality imply that

$$\|x_{k+1} - x^*\|_{\mathbf{M}} \leq \|y_k - x^*\|_{\mathbf{M}} + \|x_{k+1} - y_k\|_{\mathbf{M}}$$

$$\leq \frac{A_k}{A_{k+1}} \|x_k - x^*\|_{\mathbf{M}} + \frac{a_{k+1}}{A_{k+1}} \|v_k - x^*\|_{\mathbf{M}} + \|x_{k+1} - y_k\|_{\mathbf{M}} .$$

Rearranging and applying recursively yields that

$$A_{k+1} \|x_{k+1} - x^*\|_{\mathbf{M}} \leq A_k \|x_k - x^*\|_{\mathbf{M}} + a_{k+1} \|v_k - x^*\|_{\mathbf{M}} + A_{k+1} \|x_{k+1} - y_k\|_{\mathbf{M}}$$

$$\leq A_0 \|x_0 - x^*\|_{\mathbf{M}} + \sum_{i=0}^{k} a_{i+1} \|v_i - x^*\|_{\mathbf{M}} + \sum_{i=0}^{k} A_{i+1} \|x_{i+1} - y_i\|_{\mathbf{M}} .$$

Now, using $A_{k+1} = A_0 + \sum_{i=0}^{k} a_{i+1}$, $x_0 = v_0$, the previously-derived $\|v_i - x^*\|_{\mathbf{M}} \leq \sqrt{2p_0}$, and Cauchy-Schwarz,

$$\|x_{k+1} - x_*\|_{\mathbf{M}} \leq \sqrt{2p_0} + \frac{1}{A_{k+1}} \sqrt{\left( \sum_{i=0}^{k} \lambda_{i+1} A_{i+1} \right) \left( \sum_{i=0}^{k} \frac{A_{i+1}}{\lambda_{i+1}} \|x_{i+1} - y_i\|_{\mathbf{M}}^2 \right)} .$$

Now, since $\lambda_{k+1} A_{k+1} = a_{k+1}^2$ and $\sqrt{a + b} \leq \sqrt{a} + \sqrt{b}$ for nonnegative $a, b$ we have

$$\sqrt{\sum_{i=0}^{k} \lambda_{i+1} A_{i+1}} \leq \sum_{i=0}^{k} \sqrt{\lambda_{i+1} A_{i+1}} = \sum_{i=0}^{k} a_{i+1} = A_{k+1},$$

and the result follows from

$$\sum_{i=0}^{k} \frac{A_{i+1}}{\lambda_{i+1}} \|x_{i+1} - y_i\|_{\mathbf{M}}^2 \leq (1 - \sigma^2)^{-1} 2p_0$$

(due to Lemma 26), and $\sqrt{(1 - \sigma^2)^{-1}} \leq (1 - \sigma)^{-1}$. $\square$

We next give a basic helper lemma which will be useful in the proof of Proposition 3.

**Lemma 28.** *Let $\{B_k\}_{k \in \mathbb{N}}$ be a nonnegative, nondecreasing sequence such that $B_k \geq \sum_{i \in [k]} \alpha B_i$ for some $\alpha \in [0, 1)$ and all $k$. Then for all $k$, $B_k \geq \exp(\alpha(k - 1))B_1$.*

*Proof.* Extend $C(t) \stackrel{\text{def}}{=} B_{\lceil t \rceil}$ for all $t \geq 1$, and let $C(t) \stackrel{\text{def}}{=} \exp(\alpha(t-1))B_1$ for $t \in [0,1]$. Then for all $t \geq 1$,

$$C(t) = B_{\lceil t \rceil} \geq \alpha \sum_{i \in [\lceil t \rceil]} B_i \geq \alpha \int_0^t C(s) ds,$$

and it is easy to check that this inequality holds with equality for $t \in [0,1]$ as well. Letting $L(t)$ solve this integral inequality, i.e., $L(t) = C(t)$ for $t \in [0,1]$ and

$$L(t) = \alpha \int_0^t L(s) ds,$$

$L(t) = \exp(\alpha(t-1))C(1)$, and inequality $C(t) \geq L(t)$ yields the claim, recalling $B_k = C(k)$ for $k \in \mathbb{N}$. $\qquad\square$

Now we are ready to put everything together and prove the main result of this section.

**Proposition 3.** *Let differentiable $f$ be strictly convex, $\|x_0 - x^*\|_{\mathbf{M}} \leq R$ and $f(x_0) - f(x^*) \leq \epsilon_0$. Set $A_0 = R^2/(2\epsilon_0)$ and suppose that for some $r > 0$ the iterates of Algorithm 1 satisfy $\|x_{k+1} - y_k\|_{\mathbf{M}} \geq r$ for all $k \geq 0$. Then, the iterates also satisfy $f(x_k) - f(x^*) \leq 2\epsilon_0 \exp(-(\frac{r(1-\sigma)}{R})^{2/3}(k-1))$.*

*Proof.* First, we will show the bound

$$f(x_k) - f(x^*) \leq \frac{p_0}{A_1} \exp\left(-\frac{3}{2}\left(\frac{r(1-\sigma)}{\sqrt{p_0}}\right)^{2/3}(k-1)\right). \tag{8}$$

The reverse Hölder inequality with $p = 3/2$ states that for all $u, v \in \mathbb{R}_{>0}^k$,

$$\langle u, v \rangle \geq \left(\sum_{i \in [k]} u_i^{2/3}\right)^{3/2} \cdot \left(\sum_{i \in [k]} v_i^{-2}\right)^{-1/2}. \tag{9}$$

Lemma 23 gives $\sqrt{A_k} \geq \frac{1}{2} \sum_{i \in [k]} \sqrt{\lambda_i}$. Moreover, $\|x_i - y_{i-1}\|_{\mathbf{M}} > 0$ by the assumptions of this proposition, which implies by Lemma 24 that $A_i \geq \lambda_i > 0$ as well. Thus, we can apply (9) with $u_i = \sqrt{A_i}\|x_i - y_{i-1}\|_{\mathbf{M}}$ and $v_i = \sqrt{\lambda_i}/u_i$, yielding

$$\sqrt{A_k} \geq \frac{1}{2} \sum_{i \in [k]} \sqrt{\lambda_i} \geq \frac{1}{2}\left(\sum_{i \in [k]}\left(\sqrt{A_i}\|x_i - y_{i-1}\|_{\mathbf{M}}\right)^{2/3}\right)^{3/2}\left(\sum_{i \in [k]}\left(\frac{\sqrt{\lambda_i}}{\sqrt{A_i}\|x_i - y_{i-1}\|_{\mathbf{M}}}\right)^{-2}\right)^{-1/2}. \tag{10}$$

Applying Lemma 26 yields that

$$\sum_{i \in [k]}\left(\frac{\sqrt{\lambda_i}}{\sqrt{A_i}\|x_i - y_{i-1}\|_{\mathbf{M}}}\right)^{-2} = \sum_{i \in [k]} \frac{A_i\|x_i - y_{i-1}\|_{\mathbf{M}}^2}{\lambda_i} \leq \left(\frac{2}{1-\sigma^2}\right)p_0. \tag{11}$$

Now, since $\|x_i - y_{i-1}\|_{\mathbf{M}} \geq r$ by assumption, combining (10) and (11) gives

$$A_k^{1/3} \geq \left(\frac{1}{2}\right)^{2/3}\left(\sum_{i \in [k]} A_i^{1/3} r^{2/3}\right)\left(\left(\frac{2}{1-\sigma^2}\right)p_0\right)^{-1/3} = \sum_{i \in [k]} A_i^{1/3}\left(\frac{r^2(1-\sigma^2)}{8p_0}\right)^{1/3}.$$

Finally, applying Lemma 28 implies that for all $k \geq 0$

$$A_k \geq \exp\left(\frac{3}{2}\left(\frac{r^2(1-\sigma^2)}{p_0}\right)^{1/3}(k-1)\right)A_1.$$

Now, (8) follows from $\epsilon_k \leq p_k/A_k \leq p_0/A_k$ (we have $p_k \leq p_0$ from Lemma 26) and $(1-\sigma^2) \leq (1-\sigma)^2$. Now, by our choice of $A_0 = R^2/2\epsilon_0$, we have $p_0 = R^2$. As $A_1 \geq A_0$,

$$\frac{p_0}{A_1} \leq \frac{R^2}{A_0} = 2\epsilon_0.$$

Combining these bounds in the context of (8), and using $3/2 > 1$, yields the result. $\qquad\square$

## C   MS oracle implementation proofs

First, we prove our characterization of the optimizer of a ball-constrained problem.

**Lemma 4.** *Let $f : \mathbb{R}^d \to \mathbb{R}$ be continuously differentiable and strictly convex. For all $y \in \mathbb{R}^d$, $z = \arg\min_{z' \in \mathcal{B}_r(y)} f(z')$ either globally minimizes $f$, or $\|z - y\|_{\mathbf{M}} = r$ and $\nabla f(z) = -\frac{\|\nabla f(z)\|_{\mathbf{M}^\dagger}}{r} \mathbf{M}(z - y)$.*

*Proof.* By considering the optimality conditions of the Lagrange dual problem

$$\min_z \max_{\lambda \geq 0} f(z) + \frac{\lambda}{2} \left( \|z - y\|_{\mathbf{M}}^2 - r^2 \right),$$

we see there is some $\lambda \geq 0$ such that

$$\nabla f(z) = -\lambda \nabla_z \left( \frac{1}{2} \|z - y\|_{\mathbf{M}}^2 - \frac{r^2}{2} \right) = -\lambda \mathbf{M}(z - y).$$

If $\lambda = 0$ then $\nabla f(z) = 0$ and $z$ is a minimizer of $f$. On the other hand, if $\lambda > 0$, then $\|z - y\|_{\mathbf{M}} = r$ and $\nabla f(z) = -\lambda \mathbf{M}(z - y)$. By taking the $\mathbf{M}^\dagger$ seminorm of both sides of this condition, $\|\nabla f(z)\|_{\mathbf{M}^\dagger} = \lambda \|z - y\|_{\mathbf{M}} = \lambda r$; solving for $\lambda$ and substituting yields the result. $\qquad\square$

Next, on the path to proving Proposition 5, we give a helper result which bounds the change in the solution to a ball-constrained problem as we move the center.

**Lemma 29.** *For strictly convex, twice differentiable $f : \mathbb{R}^d \to \mathbb{R}$, let $\mathbf{M}$ be a positive semidefinite matrix where $\nabla f(u) \in \mathrm{Im}(\mathbf{M})$ for all $u \in \mathbb{R}^d$. Let $x, v \in \mathbb{R}^d$ be arbitrary vectors, and for all $t \in [0, 1]$, let*

$$y_t \overset{\text{def}}{=} tx + (1 - t)v, \; z_t \overset{\text{def}}{=} \arg\min_{z \in \mathcal{B}_r(y_t)} f(z).$$

*Then, for all $t \in [0, 1]$ we have*

$$\left\| \frac{d}{dt} z_t \right\|_{\nabla^2 f(z_t)} = \left\| \frac{d}{dt} \nabla f(z_t) \right\|_{(\nabla^2 f(z_t))^{-1}} \leq \|x - v\|_{\nabla^2 f(z_t)}.$$

*Proof.* Let $t \in [0, 1]$ be arbitrary. If $\|z_t - y_t\|_{\mathbf{M}} < r$, then $z_t$ is the minimizer of $f$, i.e. $\nabla f(z_t) = 0$ and $\frac{d}{dt} z_t = 0$ yielding the result (as in this case the minimizer stays in the interior for small perturbations of $y_t$). For the remainder of the proof assume that $\|z_t - y_t\|_{\mathbf{M}} = r$, in which case Lemma 4 yields that

$$\nabla f(z_t) = -\frac{\|\nabla f(z_t)\|_{\mathbf{M}^\dagger}}{r} \mathbf{M}(z_t - y_t). \tag{12}$$

Now, differentiating both sides with respect to $t$ yields that

$$\frac{d}{dt} (\nabla f(z_t)) = -\frac{\left\langle \nabla f(z_t), \mathbf{M}^\dagger \frac{d}{dt} (\nabla f(z_t)) \right\rangle}{r \|\nabla f(z_t)\|_{\mathbf{M}^\dagger}} \mathbf{M}(z_t - y_t) - \frac{1}{r} \|\nabla f(z_t)\|_{\mathbf{M}^\dagger} \mathbf{M} \left( \frac{d}{dt} z_t - (x - v) \right). \tag{13}$$

Combining (12) and (13) and taking an inner product of both sides with $\mathbf{M}^\dagger \frac{d}{dt}(\nabla f(z_t))$ yields that

$$\left\| \frac{d}{dt} (\nabla f(z_t)) \right\|_{\mathbf{M}^\dagger}^2 = \frac{\left\langle \nabla f(z_t), \mathbf{M}^\dagger \frac{d}{dt}(\nabla f(z_t)) \right\rangle^2}{\|\nabla f(z_t)\|_{\mathbf{M}^\dagger}^2} - \frac{1}{r} \|\nabla f(z_t)\|_{\mathbf{M}^\dagger} \left\langle \frac{d}{dt} z_t - (x - v), \mathbf{M}\mathbf{M}^\dagger \frac{d}{dt}(\nabla f(z_t)) \right\rangle.$$

Next, Cauchy-Schwarz implies $\left\langle \nabla f(z_t), \mathbf{M}^\dagger \frac{d}{dt}(\nabla f(z_t)) \right\rangle^2 \leq \|\nabla f(z_t)\|_{\mathbf{M}^\dagger}^2 \cdot \|\frac{d}{dt}(\nabla f(z_t))\|_{\mathbf{M}^\dagger}^2$, so the first two terms in the above display cancel. Rearranging the last term yields

$$\left\langle \frac{d}{dt} z_t, \mathbf{M}\mathbf{M}^\dagger \frac{d}{dt}(\nabla f(z_t)) \right\rangle \leq \left\langle x - v, \mathbf{M}\mathbf{M}^\dagger \frac{d}{dt}(\nabla f(z_t)) \right\rangle.$$

Since $\nabla f(z_t)$ is in the image of $\mathbf{M}$ for all $t$, $\frac{d}{dt}(\nabla f(z_t))$ must also be in the image of $\mathbf{M}$. Thus, we can drop the $\mathbf{M}\mathbf{M}^\dagger$ matrices from the above expression. Also as $\frac{d}{dt}(\nabla f(z_t)) = \nabla^2 f(z_t) \frac{d}{dt} z_t$, this simplifies to

$$\left\| \frac{d}{dt} z_t \right\|_{\nabla^2 f(z_t)}^2 \leq \left\langle x - v, \nabla^2 f(z_t) \frac{d}{dt} z_t \right\rangle \leq \left\| \frac{d}{dt} z_t \right\|_{\nabla^2 f(z_t)} \cdot \|x - v\|_{\nabla^2 f(z_t)}.$$

Dividing both sides by $\|\frac{d}{dt}z_t\|_{\nabla^2 f(z_t)}$ and applying $\frac{d}{dt}\nabla f(z_t) = \nabla^2 f(z_t)\frac{d}{dt}z_t$ then yields the result. $\qquad\square$

We now bound the Lipschitz constant of the function $g(\lambda) = \lambda\,\|\nabla f(z_{t_\lambda})\|_{\mathbf{M}^\dagger}$, where we recall the definitions

$$a_\lambda \stackrel{\text{def}}{=} \frac{\lambda + \sqrt{\lambda^2 + 4\lambda A}}{2}, \ t_\lambda \stackrel{\text{def}}{=} \frac{A}{A + a_\lambda}, \ y_{t_\lambda} \stackrel{\text{def}}{=} t_\lambda x + (1 - t_\lambda)v, \ z_{t_\lambda} \stackrel{\text{def}}{=} \min_{z \in \mathcal{B}_r(y_{t_\lambda})} f(z). \quad (14)$$

**Lemma 30.** *Let $f$ be $L$-smooth in $\|\cdot\|_{\mathbf{M}}$. Assume that in (14), $\|x - x^*\|_{\mathbf{M}} \le D$ and $\|v - x^*\|_{\mathbf{M}} \le D$. For all $\lambda \ge 0$,*

$$\left| \frac{d}{d\lambda} g(\lambda) \right| \le L(2D + r).$$

*Proof.* We compute

$$\frac{d}{d\lambda}g(\lambda) = \|\nabla f(z_{t_\lambda})\|_{\mathbf{M}^\dagger} + \lambda \frac{\left\langle \nabla f(z_{t_\lambda}), \mathbf{M}^\dagger \nabla^2 f(z_{t_\lambda})\left(\frac{d}{dt_\lambda}z_{t_\lambda}\right) \right\rangle}{\|\nabla f(z_{t_\lambda})\|_{\mathbf{M}^\dagger}} \frac{d}{d\lambda}t_\lambda. \quad (15)$$

First, direct calculation yields

$$\frac{d}{d\lambda}t_\lambda = -\frac{A}{(A + a_\lambda)^2}\frac{d}{d\lambda}a_\lambda = -\frac{A}{(A + a_\lambda)^2}\cdot\frac{1}{2}\left(1 + \left(\lambda^2 + 4A\lambda\right)^{-1/2}(\lambda + 2A)\right).$$

Consequently, recalling the definition of $a_\lambda$,

$$\begin{aligned}\left|\lambda\frac{d}{d\lambda}t_\lambda\right| &= \left|\frac{2A\lambda}{(2A + \lambda + \sqrt{\lambda^2 + 4A\lambda})^2}\left(1 + \frac{\lambda + 2A}{\sqrt{\lambda^2 + 4A\lambda}}\right)\right| \\ &= \left|\frac{2A\lambda}{(2A + \lambda + \sqrt{\lambda^2 + 4A\lambda})\sqrt{\lambda^2 + 4A\lambda}}\right| \le \frac{2A\lambda}{\lambda^2 + 4A\lambda} \le \frac{1}{2}.\end{aligned} \quad (16)$$

where we used that $A, \lambda > 0$. Next, by triangle inequality and smoothness in the $\mathbf{M}$-norm,

$$\|\nabla f(z_{t_\lambda})\|_{\mathbf{M}^\dagger} \le L\|z_{t_\lambda} - x^*\|_{\mathbf{M}} \le L\left(\|z_{t_\lambda} - y_{t_\lambda}\|_{\mathbf{M}} + \|y_{t_\lambda} - x^*\|_{\mathbf{M}}\right) \le L(r + D). \quad (17)$$

In the last inequality, we used convexity of norms and $\|x - x^*\|_{\mathbf{M}}, \ \|v - x^*\|_{\mathbf{M}} \le D$. The final bound we require is due to Lemma 29: observe

$$\begin{aligned}&\left\langle \nabla f(z_{t_\lambda})\mathbf{M}^\dagger, \nabla^2 f(z_{t_\lambda})\left(\frac{d}{dt_\lambda}z_{t_\lambda}\right)\right\rangle \le \|\nabla f(z_{t_\lambda})\|_{\mathbf{M}^\dagger\nabla^2 f(z_{t_\lambda})\mathbf{M}^\dagger}\left\|\frac{d}{dt_\lambda}z_{t_\lambda}\right\|_{\nabla^2 f(z_{t_\lambda})} \\ &\le \sqrt{L}\|\nabla f(z_{t_\lambda})\|_{\mathbf{M}^\dagger}\|x - v\|_{\nabla^2 f(z_{t_\lambda})} \le L\|\nabla f(z_{t_\lambda})\|_{\mathbf{M}^\dagger}\|x - v\|_{\mathbf{M}} \le 2LD\|\nabla f(z_{t_\lambda})\|_{\mathbf{M}^\dagger}.\end{aligned} \quad (18)$$

The first inequality is by Cauchy-Schwarz, the second is due to Lemma 29 and $\mathbf{M}^\dagger\nabla^2 f(z_{t_\lambda})\mathbf{M}^\dagger \preceq L\mathbf{M}^\dagger$ by smoothness, and the third is again from smoothness with $\nabla^2 f(z_{t_\lambda}) \preceq L\mathbf{M}$. Combining (15), (16), (17), and (18) yields the claim. $\qquad\square$

We now prove Proposition 5.

**Proposition 5** (Guarantees of Algorithm 2). *Let $L, D, \delta, r > 0$ and $\mathcal{O}_{\text{ball}}$ satisfy the requirements in Lines 1–3 of Algorithm 2, and $\epsilon < 2LD^2$. Then, Algorithm 2 either returns $\tilde{z}_{t_\lambda}$ with $f(\tilde{z}_{t_\lambda}) - f(x^*) < \epsilon$, or implements a $\frac{1}{2}$-MS oracle with the additional guarantee $\|\tilde{z}_{t_\lambda} - y_{t_\lambda}\|_{\mathbf{M}} \ge \frac{11r}{12}$. Moreover, the number of calls to $\mathcal{O}_{\text{ball}}$ is bounded by $O(\log(\frac{LD^2}{\epsilon}))$.*

*Proof.* This proof will require three bounds on the size of the parameter $\delta$ used in the ball optimization oracle. We state them here, and show that the third implies the other two. We require

$$\delta \le \min\left\{ \frac{\epsilon}{2L(D + r)}, \ \sqrt{\frac{2\epsilon}{L}}, \ \frac{r}{12\left(1 + \frac{2L(D+r)r}{\epsilon}\right)} \right\}. \quad (19)$$

The fact that the third bound implies the first is clear, and the second is implied by the assumption $2LD^2 > \epsilon$.

Our goal is to first show that if $g(u) > r$, then we have an $\epsilon$-approximate minimizer; otherwise, we construct a range $[\ell, u]$ which contains some $\lambda$ with $g(\lambda) = r$, and we apply the Lipschitz condition Lemma 30 to prove correctness of our binary search. Recall that for every $\lambda$, the guarantees of $\mathcal{O}_{\text{ball}}$ imply that $\|z_{t_\lambda} - \tilde{z}_{t_\lambda}\|_{\mathbf{M}} \leq \delta$, and moreover

$$\|\tilde{z}_{t_\lambda} - x^*\|_{\mathbf{M}} \leq \|\tilde{z}_{t_\lambda} - y_{t_\lambda}\|_{\mathbf{M}} + \|y_{t_\lambda} - x^*\|_{\mathbf{M}} \leq D + r$$

by convexity. Thus, if it holds that $\|\nabla f(\tilde{z}_{t_u})\|_{\mathbf{M}^\dagger} \leq r/u + L\delta$ in Line 7, then

$$f(\tilde{z}_{t_u}) - f(x^*) \leq \langle \nabla f(\tilde{z}_{t_u}), \tilde{z}_{t_u} - x^* \rangle \leq \|\nabla f(\tilde{z}_{t_u})\|_{\mathbf{M}^\dagger}(D + r) \leq \epsilon,$$

for our choice of $u = 2(D + r)r/\epsilon$ and $\delta \leq \epsilon/(2L(D + r))$ (19). On the other hand, if $\|\nabla f(\tilde{z}_{t_u})\|_{\mathbf{M}^\dagger} \geq r/u + L\delta$, by Lipschitzness of the gradient and the guarantee $\|\tilde{z}_{t_\lambda} - z_{t_\lambda}\|_{\mathbf{M}} \leq \delta$, we have $g(u) = u\|\nabla f(z_{t_u})\|_{\mathbf{M}^\dagger} \geq r$. Moreover, for $\ell = r/L(D + r)$, by Lipschitzness of the gradient from $x^*$,

$$g(\ell) = \ell\|\nabla f(z_{t_\ell})\|_{\mathbf{M}^\dagger} \leq \frac{r}{L(D + r)}(L(D + r)) \leq r.$$

By continuity, it is clear that for some value $\lambda \in [\ell, u]$, $g(\lambda) = r$; we note the assumption $2LD^2 > \epsilon$ guarantees that $\ell < u$, so the search range is valid. Next, if for some value of $\lambda$, $z_{t_\lambda} = x^*$, as long as $\delta \leq \sqrt{2\epsilon/L}$, we have by smoothness

$$f(\tilde{z}_{t_\lambda}) - f(x^*) = f(\tilde{z}_{t_\lambda}) - f(z_{t_\lambda}) \leq \frac{L\delta^2}{2} \leq \epsilon.$$

Otherwise, $z_{t_\lambda}$ is on the boundary of the ball around $y_{t_\lambda}$, so that we have the desired

$$\|\tilde{z}_{t_\lambda} - y_{t_\lambda}\|_{\mathbf{M}} \geq r - \delta \geq \frac{11r}{12}.$$

Moreover, (4) implies

$$\begin{aligned}
\left\|\tilde{z}_{t_\lambda} - (y_{t_\lambda} - \lambda\mathbf{M}^\dagger\nabla f(\tilde{z}_{t_\lambda}))\right\|_{\mathbf{M}} &\leq (1 + L\lambda)\delta + \left\|z_{t_\lambda} - (y_{t_\lambda} - \lambda\mathbf{M}^\dagger\nabla f(z_{t_\lambda}))\right\|_{\mathbf{M}} \\
&= (1 + L\lambda)\delta + \left\|\left(\lambda - \frac{r}{\|\nabla f(z_{t_\lambda})\|_{\mathbf{M}^\dagger}}\right)\mathbf{M}^\dagger\nabla f(z_{t_\lambda})\right\|_{\mathbf{M}} \\
&= (1 + L\lambda)\delta + \left\|\left(\frac{g(\lambda) - r}{\|\nabla f(z_{t_\lambda})\|_{\mathbf{M}^\dagger}}\right)\mathbf{M}^\dagger\nabla f(z_{t_\lambda})\right\|_{\mathbf{M}} \\
&\leq (1 + Lu)\delta + |g(\lambda) - r|.
\end{aligned}$$

So, as long as $\delta \leq r/(12(1 + Lu))$ and $|g(\lambda) - r| \leq r/4$, we have the desired $\frac{1}{2}$-MS oracle guarantee

$$\begin{aligned}
\left\|\tilde{z}_{t_\lambda} - (y_{t_\lambda} - \lambda\mathbf{M}^\dagger\nabla f(\tilde{z}_{t_\lambda}))\right\|_{\mathbf{M}} &\leq \frac{r}{12} + \frac{r}{4} \leq \frac{1}{2}(r - \delta) \\
&\leq \frac{1}{2}\|\tilde{z}_{t_\lambda} - y_{t_\lambda}\|_{\mathbf{M}}.
\end{aligned}$$

Thus, the algorithm can terminate whenever we can guarantee $|g(\lambda) - r| \leq r/4$. We can certify the value of $g(\lambda)$ via $\lambda\|\nabla f(\tilde{z}_{t_\lambda})\|_{\mathbf{M}^\dagger}$ up to additive error $L\lambda\delta \leq r/12$, so that $|\lambda\|\nabla f(\tilde{z}_{t_\lambda})\|_{\mathbf{M}^\dagger} - r| \leq r/6$ implies $|g(\lambda) - r| \leq r/4$. Finally, let $\lambda^*$ be any value in $[\ell, u]$ where $g(\lambda^*) = r$. By Lemma 30,

$$|\lambda - \lambda^*| \leq \frac{r}{12(L(2D + r))} \implies |g(\lambda) - r| \leq \frac{r}{12}$$

$$\implies |\lambda\|\nabla f(\tilde{z}_{t_\lambda})\|_{\mathbf{M}^\dagger} - r| \leq r/6, \text{ i.e. search terminates.}$$

In conclusion, we can bound the number of calls required by Algorithm 2 in executions of Lines 16 and 20 to $\mathcal{O}_{\text{ball}}$ by

$$\log\left((u - \ell) \cdot \left(\frac{r}{12(L(2D + r))}\right)^{-1}\right) \leq \log\left(\frac{4Dr}{\epsilon} \cdot \frac{36LD}{r}\right).$$

$\square$

**Theorem 6** (Acceleration with a ball optimization oracle). *Let $\mathcal{O}_{\text{ball}}$ be an $(\frac{r}{12+126LRr/\epsilon}, r)$-ball optimization oracle for strictly convex and $L$-smooth $f : \mathbb{R}^d \to \mathbb{R}$ with minimizer $x^*$, and initial point $x_0$ satisfying $\|x_0 - x^*\|_{\mathbf{M}} \leq R$ and $f(x_0) - f(x^*) \leq \epsilon_0$. Then, Algorithm 1 using Algorithm 2 as a Monteiro-Svaiter oracle with $D = \sqrt{18}R$ produces an iterate $x_k$ with $f(x_k) - f(x^*) \leq \epsilon$, in $O\left((R/r)^{2/3} \log(\epsilon_0/\epsilon) \log(LR^2/\epsilon)\right)$ calls to $\mathcal{O}_{\text{ball}}$.*

*Proof.* More specifically, we will return the point encountered in Algorithm 1 with the smallest function value, in the case Proposition 5 ever guarantees a point is an $\epsilon$-approximate minimizer. Note that Lemma 27 implies that in each run of Algorithm 2, it suffices to set $D = 3\sqrt{2}R$, where we recall (in its context) $\sqrt{2p_0} = \sqrt{2}R$, via the proof of Proposition 3. Recalling $R > r$, this implies that setting

$$\delta = \frac{r}{12(1 + \frac{2L(D+r)r}{\epsilon})} \geq \frac{r}{12 + \frac{126LRr}{\epsilon}}$$

suffices in the guarantees of $\mathcal{O}_{\text{ball}}$. Moreover, if the assumption $\epsilon < 2LD^2$ in Proposition 5 does not hold, smoothness implies we may return any $x_k$. The oracle complexity follows by combining Proposition 5 with Proposition 3. $\square$

# D   Algorithm and Proofs for Theorem 8

We prove Theorem 8 in two parts. First, we provide a convergence guarantee for trust region subproblems, and then use it as a primitive in Algorithm 7, an accelerated ball-constrained Newton's method. Finally, we describe a sufficient condition for Hessian stability to hold.

## D.1   Trust region subproblems

We describe a procedure for solving the convex trust region problem

$$\min_{x \in \mathcal{B}_r(\bar{x})} Q(x) \stackrel{\text{def}}{=} -g^\top x + \frac{1}{2}x^\top \mathbf{H}x.$$

While trust region problems of this form are well-studied [15, 21], we could not find a concrete bound on the number of linear system solutions required to solve them approximately. Here we describe the procedure $\text{SOLVETR}(\bar{x}, r, g, \mathbf{H}, \mathbf{M}, \Delta)$ (Algorithm 6) that uses a well-known binary search strategy to solve the trust-region problem to accuracy $\Delta$. The procedure enjoys the convergence guarantee as stated in Proposition 34.

---

**Algorithm 6** $\text{SOLVETR}(\bar{x}, r, g, \mathbf{H}, \mathbf{M}, \Delta)$

---

1: Let $0 < \mu \leq L$ so $\mu\mathbf{M} \preceq \mathbf{H} \preceq L\mathbf{M}$, and let $\Delta > 0$.
2: $\hat{g} \leftarrow g - \mathbf{H}\bar{x}$
3: $\ell \leftarrow 0$, $u \leftarrow \frac{\|\hat{g}\|_{\mathbf{M}^\dagger}}{r}$, $\iota \leftarrow \frac{\Delta\mu^2}{\|\hat{g}\|_{\mathbf{M}^\dagger}}$
4: **if** $\|\mathbf{H}^\dagger \hat{g}\|_{\mathbf{M}} \leq r$ **then**
5:    **return** $\mathbf{H}^\dagger \hat{g}$
6: **else**
7:    $\lambda \leftarrow \frac{\ell+u}{2}$, $\lambda^- \leftarrow \lambda - \iota$
8:    **while not** $(\|(\mathbf{H}+\lambda\mathbf{M})^\dagger \hat{g}\|_{\mathbf{M}} \leq r)$ **and** $(r < \|(\mathbf{H}+\lambda^-\mathbf{M})^\dagger \hat{g}\|_{\mathbf{M}})$ **do**
9:       **if** $\|(\mathbf{H}+\lambda\mathbf{M})^\dagger \hat{g}\|_{\mathbf{M}} \leq r$ **then**
10:          $u \leftarrow \lambda$, $\lambda \leftarrow \frac{\ell+u}{2}$, $\lambda^- \leftarrow \lambda - \iota$
11:       **else**
12:          $\ell \leftarrow \lambda$, $\lambda \leftarrow \frac{\ell+u}{2}$, $\lambda^- \leftarrow \lambda - \iota$
13:       **end if**
14:    **end while**
15:    **return** $(\mathbf{H}+\lambda\mathbf{M})^\dagger \hat{g}$
16: **end if**

---

For simplicity, we first focus on developing technical results for the trust region problem of the following form (below $\mathbf{0}$ is the origin)

$$\min_{x \in \mathcal{B}_r(\mathbf{0})} Q(x) \stackrel{\text{def}}{=} -g^\top x + \frac{1}{2}x^\top \mathbf{H}x; \tag{20}$$

our final guarantees will be obtained by an appropriate linear shift. All results in this section assume $\mu\mathbf{M} \preceq \mathbf{H} \preceq L\mathbf{M}$ for some $0 < \mu \le L$, which in particular implies that $\mathbf{H}$ and $\mathbf{M}$ share a kernel. We first state a helpful monotonicity property which will be used throughout.

**Lemma 31.** $\left\|(\mathbf{H} + \lambda\mathbf{M})^\dagger g\right\|_{\mathbf{M}}$ *is monotonically decreasing in* $\lambda$, *for any vector g.*

*Proof.* We will refer to the projection onto the column space of $\mathbf{M}$, i.e. $\mathbf{M}\mathbf{M}^\dagger$, by $\tilde{\mathbf{I}}$. To show the lemma, it suffices to prove that

$$(\mathbf{H} + \lambda\mathbf{M})^\dagger \mathbf{M}(\mathbf{H} + \lambda\mathbf{M})^\dagger$$

is monotone in the Loewner order. Denoting $\tilde{\mathbf{H}} \overset{\text{def}}{=} \mathbf{M}^{\dagger/2}\mathbf{H}\mathbf{M}^{\dagger/2}$,

$$(\mathbf{H} + \lambda\mathbf{M})^\dagger = \left(\mathbf{M}^{1/2}\left(\tilde{\mathbf{H}} + \lambda\tilde{\mathbf{I}}\right)\mathbf{M}^{1/2}\right)^\dagger = \mathbf{M}^{\dagger/2}\left(\tilde{\mathbf{H}} + \lambda\tilde{\mathbf{I}}\right)^\dagger \mathbf{M}^{\dagger/2}. \tag{21}$$

Therefore, it suffices to show that

$$\mathbf{M}^{\dagger/2}\left(\tilde{\mathbf{H}} + \lambda\tilde{\mathbf{I}}\right)^\dagger \left(\tilde{\mathbf{H}} + \lambda\tilde{\mathbf{I}}\right)^\dagger \mathbf{M}^{\dagger/2}$$

is monotone in the Lowener order, which follows as $\tilde{\mathbf{H}}$ and $\tilde{\mathbf{I}}$ commute. $\qquad\square$

Next, we characterize the minimizer to (20).

**Lemma 32.** *A solution to (20) is given by* $x_{g,\mathbf{H}} = (\mathbf{H} + \lambda\mathbf{M})^\dagger g$ *for a unique value of* $\lambda \ge 0$. *Unless* $\lambda = 0$, $\|x_{g,\mathbf{H}}\|_{\mathbf{M}} = r$.

*Proof.* By considering the optimality conditions of the Lagrange dual problem

$$\min_x \max_{\lambda \ge 0} -g^\top x + \frac{1}{2}x^\top \mathbf{H}x + \frac{\lambda}{2}\left(x^\top \mathbf{M}x - r^2\right),$$

either $\lambda = 0$ and the minimizer $\mathbf{H}^\dagger g$ is in $\mathcal{B}_r(0)$, or there is $x_{g,\mathbf{H}} = (\mathbf{H} + \lambda\mathbf{M})^\dagger g$ on the region boundary (linear shifts in the kernel of $\mathbf{M}$ do not affect the $\mathbf{M}$ norm constraint or the objective, so we may restrict to the column space without loss of generality). Uniqueness of $\lambda$ then follows from Lemma 31. $\qquad\square$

Next, we bound how tightly we must approximate the value $\lambda$ in order to obtain an approximate minimizer to (D.1).

**Lemma 33.** *Suppose* $g \in \text{Im}(\mathbf{M})$, *and* $\left\|\mathbf{H}^\dagger g\right\|_{\mathbf{M}} > r$. *Then, for* $\lambda^* > 0$ *such that* $\left\|(\mathbf{H} + \lambda^*\mathbf{M})^\dagger g\right\|_{\mathbf{M}} = r$, *and any* $\lambda > 0$ *such that* $|\lambda - \lambda^*| \le \frac{\Delta\mu^2}{\|g\|_{\mathbf{M}^\dagger}}$, *we have*

$$\left\|(\mathbf{H} + \lambda\mathbf{M})^\dagger g - x_{g,\mathbf{H}}\right\|_{\mathbf{M}} \le \Delta. \tag{22}$$

*Proof.* We follow the notation of Lemma 31. Recalling (21), we expand

$$\left\|(\mathbf{H} + \lambda\mathbf{M})^\dagger g - x_{g,\mathbf{H}}\right\|_{\mathbf{M}}^2 = \tilde{g}^\top \left(\left(\tilde{\mathbf{H}} + \lambda\tilde{\mathbf{I}}\right)^\dagger - \left(\tilde{\mathbf{H}} + \lambda^*\tilde{\mathbf{I}}\right)^\dagger\right)^2 \tilde{g}. \tag{23}$$

Here, we defined $\tilde{g} = \mathbf{M}^{\dagger/2}g$. Note that $\|\tilde{g}\|_2^2 = \|g\|_{\mathbf{M}^\dagger}^2$, where we used $g \in \text{Im}(\mathbf{M})$. Without loss of generality, since $\tilde{\mathbf{H}} + \lambda\tilde{\mathbf{I}}$ commute for all $\lambda$ therefore simultaneously diagonalizable, suppose we are in the basis where $\tilde{\mathbf{H}}$ is diagonal and has diagonal entries $\{h_i\}_{i \in [d]}$. Expanding the right hand side of (23), we have

$$\sum_{i \in [d]} \tilde{g}_i^2 \left(\frac{1}{h_i + \lambda} - \frac{1}{h_i + \lambda^*}\right)^2 = \sum_{i \in [d]} \tilde{g}_i^2 \left(\frac{(\lambda^* - \lambda)^2}{(h_i + \lambda)^2(h_i + \lambda^*)^2}\right)$$

$$\le \sum_{i \in [d]} \frac{\tilde{g}_i^2}{\mu^4}\left(\frac{\Delta\mu^2}{\|\tilde{g}\|_{\mathbf{M}^\dagger}}\right)^2 \le \Delta^2.$$

In the last inequality, note that whenever $h_i \neq 0$, it is at least $\mu$ by strong convexity in $\|\cdot\|_{\mathbf{M}}$, and whenever $h_i$ is zero, so is $\tilde{g}_i$, by the assumption on $g$ and the fact that $\mathbf{M}$ and $\mathbf{H}$ share a kernel. $\qquad\square$

Finally, by combining these building blocks, we obtain a procedure for solving (D.1) to high accuracy.

**Proposition 34.** *Let $\mathbf{M}$ and $\mathbf{H}$ share a kernel, $\mu\mathbf{M} \preceq \mathbf{H}$ for $\mu > 0$, and let $\Delta > 0$. The procedure* $\textsc{SolveTR}(\bar{x}, r, g, \mathbf{H}, \mathbf{M}, \Delta)$ *solves*

$$
O\left(\log\left(\frac{\|\mathbf{H}\bar{x} - g\|_{\mathbf{M}^\dagger}^2}{r\mu^2\Delta}\right)\right)
$$

*linear systems in matrices of the form $\mathbf{H} + \lambda\mathbf{M}$ for $\lambda \geq 0$, and returns $\tilde{x} \in \mathcal{B}_r(\bar{x})$ with $\|\tilde{x} - x_{g,\mathbf{H}}\|_{\mathbf{M}} \leq \Delta$, where*

$$
x_{g,\mathbf{H}} \in \arg\min_{x \in \mathcal{B}_r(\bar{x})} -g^\top x + \frac{1}{2}x^\top \mathbf{H}x.
$$

*Proof.* First, for $\hat{g} \overset{\text{def}}{=} g - \mathbf{H}\bar{x}$, we have the equivalent problem

$$
\arg\min_{\|y\|_{\mathbf{M}}\leq r} -g^\top(y + \bar{x}) + \frac{1}{2}(y + \bar{x})^\top\mathbf{H}(y + \bar{x}) = \arg\min_{y \in \mathcal{B}_r(0)} -\hat{g}^\top y + \frac{1}{2}y^\top\mathbf{H}y.
$$

Following Lemma 32, in Line 4 we verify whether for the optimal solution, $\lambda = 0$, using one linear system solve. If not, by monotonicity of $\left\|(\mathbf{H} + \lambda\mathbf{M})^\dagger\hat{g}\right\|_{\mathbf{M}}$ in $\lambda$ (Lemma 31), it is clear that the value $\lambda^*$ corresponding to the solution lies in the range $[\ell, u] = [0, \|\hat{g}\|_{\mathbf{M}^\dagger}/r]$, by

$$
\left\|\left(\mathbf{H} + \frac{\|\hat{g}\|_{\mathbf{M}^\dagger}}{r}\mathbf{M}\right)^\dagger\hat{g}\right\|_{\mathbf{M}}^2 \leq r^2.
$$

This follows from e.g. the characterization (21). Therefore, Lemma 33 shows that it suffices to perform a binary search over this region to find a value $\lambda$ with additive error $\iota = \frac{\Delta\mu^2}{\|\hat{g}\|_{\mathbf{M}^\dagger}}$ to output a solution $\tilde{x}$ of the desired accuracy. We note that we may check feasibility in $\mathcal{B}_r(0)$ by computing the value of $\left\|(\mathbf{H} + \lambda\mathbf{M})^\dagger\hat{g}\right\|_{\mathbf{M}}$ due to Lemma 32, and it suffices to output the larger value of $\lambda$ amongst the endpoint of the interval of length $\iota$ containing $\lambda^*$, reflecting our termination condition in Line 8. $\qquad\square$

## D.2 Accelerated Newton method

Theorem 8 follows from an analysis of Algorithm 7, which is essentially Nesterov's accelerated gradient method in the Euclidean seminorm $\|\cdot\|_{\mathbf{H}}$ with $\mathbf{H} = \nabla^2 f(\bar{x})$, or equivalently a sequence of constrained Newton steps using the Hessian of the center point $\bar{x}$. Other works [16, 14] consider variants of Nesterov's accelerated method in arbitrary norms and under various noise assumptions, but do not give convergence guarantees compatible with the type of error incurred by our trust region subproblem solver. We state the convergence guarantee below, and defer its proof to Appendix D.2 for completeness; it is a simple adaptation of the standard acceleration analysis under inexact subproblem solves.

---

**Algorithm 7** Accelerated Newton's method

---

1: **Input:** Radius $r$ and accuracy $\delta$ such that $r \geq \delta > 0$.
2: **Input:** Function $f : \mathbb{R}^d \to \mathbb{R}$ that is $L$-smooth, $\mu$-strongly convex, and $(r, c)$-Hessian stable in $\|\cdot\|_{\mathbf{M}}$ with minimizer $x^*$.
3: **Input:** Center point $\bar{x} \in \mathbb{R}^d$ satisfying $\|\bar{x} - x^*\|_{\mathbf{M}} \leq D$.
4: $\mathbf{H} \leftarrow \nabla^2 f(\bar{x}),\ \alpha \leftarrow c^{-1},\ \Delta \leftarrow \frac{\mu\delta^2}{4Lc(5r+D)},\ x_0 \leftarrow \bar{x},\ z_0 \leftarrow \bar{x}$
5: **for** $k = 0, 1, 2, \ldots$ **do**
6: $\quad y_k \leftarrow \frac{1}{1+\alpha}x_k + \frac{\alpha}{1+\alpha}z_k,\ g_k \leftarrow \nabla f(y_k) - \mathbf{H}(\alpha y_k + (1 - \alpha)z_k)$
7: $\quad z_{k+1} \leftarrow \textsc{SolveTR}(\bar{x}, r, g_k, \mathbf{H}, \mathbf{M}, \Delta)$
8: $\quad x_{k+1} \leftarrow \alpha z_{k+1} + (1 - \alpha)x_k$
9: **end for**

---

This section gives the guarantees of Algorithm 7, and in particular a proof of Theorem 8. Throughout, assume $\mu\mathbf{M} \preceq \mathbf{H} \preceq L\mathbf{M}$, where $\mathbf{H} = \nabla^2 f(\bar{x})$. We note that Line 7 of Algorithm 7 is approximately

implementing the step

$$z_{k+1}^{\text{ideal}} \leftarrow \underset{z \in \mathcal{B}_r(\bar{x})}{\arg\min} \left\{ \langle \nabla f(y_k), z \rangle + \frac{1-\alpha}{2} \|z - z_k\|_{\mathbf{H}}^2 + \frac{\alpha}{2} \|z - y_k\|_{\mathbf{H}}^2 \right\}$$

$$= \underset{z \in \mathcal{B}_r(\bar{x})}{\arg\min} \left\{ \langle \nabla f(y_k) - (1-\alpha)\mathbf{H}z_k - \alpha\mathbf{H}y_k, z \rangle + \frac{1}{2} z^\top \mathbf{H} z \right\}, \tag{24}$$

with the guarantee $\left\| z_{k+1}^{\text{ideal}} - z_{k+1} \right\|_{\mathbf{M}} \leq \Delta$. Throughout, we denote $x_{\bar{x},r}$ as the minimizer of $f$ in $\mathcal{B}_r(\bar{x})$.

**Lemma 35.** *Consider a single iteration of Algorithm 7 from a pair of points $x_k, z_k$. We have*

$$f(y_k) + \langle \nabla f(y_k), z_{k+1}^{\text{ideal}} - y_k \rangle + \frac{\alpha}{2} \|y_k - z_{k+1}^{\text{ideal}}\|_{\mathbf{H}}^2 + \frac{1-\alpha}{2} \|z_k - z_{k+1}^{\text{ideal}}\|_{\mathbf{H}}^2$$

$$\leq f(x_{\bar{x},r}) + \frac{1-\alpha}{2} \|z_k - x_{\bar{x},r}\|_{\mathbf{H}}^2 - \frac{1}{2} \|z_{k+1}^{\text{ideal}} - x_{\bar{x},r}\|_{\mathbf{H}}^2.$$

*Proof.* By the first-order optimality conditions of $z_{k+1}^{\text{ideal}}$ with respect to $x_{\bar{x},r}$,

$$\langle \nabla f(y_k), z_{k+1}^{\text{ideal}} - x_{\bar{x},r} \rangle \leq \frac{1-\alpha}{2} \left( \|z_k - x_{\bar{x},r}\|_{\mathbf{H}}^2 - \|z_k - z_{k+1}^{\text{ideal}}\|_{\mathbf{H}}^2 \right)$$

$$+ \frac{\alpha}{2} \left( \|y_k - x_{\bar{x},r}\|_{\mathbf{H}}^2 - \|y_k - z_{k+1}^{\text{ideal}}\|_{\mathbf{H}}^2 \right) - \frac{1}{2} \|z_{k+1}^{\text{ideal}} - x_{\bar{x},r}\|_{\mathbf{H}}^2.$$

Here, we twice-used the well-known identity $\langle \mathbf{H}(z_{k+1}^{\text{ideal}} - x), z_{k+1}^{\text{ideal}} - x_{\bar{x},r} \rangle = \frac{1}{2} \|z_{k+1}^{\text{ideal}} - x_{\bar{x},r}\|_{\mathbf{H}}^2 + \frac{1}{2} \|z_{k+1}^{\text{ideal}} - x\|_{\mathbf{H}}^2 - \frac{1}{2} \|x - x_{\bar{x},r}\|_{\mathbf{H}}^2$. Rearranging this and using strong convexity, where we recall $\alpha = c^{-1}$,

$$f(y_k) + \langle \nabla f(y_k), z_{k+1}^{\text{ideal}} - y_k \rangle + \frac{\alpha}{2} \|y_k - z_{k+1}^{\text{ideal}}\|_{\mathbf{H}}^2 + \frac{1-\alpha}{2} \|z_k - z_{k+1}^{\text{ideal}}\|_{\mathbf{H}}^2$$

$$\leq \left( f(y_k) + \langle \nabla f(y_k), x_{\bar{x},r} - y_k \rangle + \frac{\alpha}{2} \|y_k - x_{\bar{x},r}\|_{\mathbf{H}}^2 \right) + \frac{1-\alpha}{2} \|z_k - x_{\bar{x},r}\|_{\mathbf{H}}^2 - \frac{1}{2} \|z_{k+1}^{\text{ideal}} - x_{\bar{x},r}\|_{\mathbf{H}}^2$$

$$\leq f(x_{\bar{x},r}) + \frac{1-\alpha}{2} \|z_k - x_{\bar{x},r}\|_{\mathbf{H}}^2 - \frac{1}{2} \|z_{k+1}^{\text{ideal}} - x_{\bar{x},r}\|_{\mathbf{H}}^2.$$

$\square$

Next, we modify the guarantee of Lemma 35 to tolerate an inexact step on the point $z_{k+1}$. We use the following lemma.

**Lemma 36.** *Suppose the convex function $h$ is $L$-smooth in $\|\cdot\|_{\mathbf{M}}$ in a region $\mathcal{X}$ with bounded $\|\cdot\|_{\mathbf{M}}$ diameter $2r$, and $x_h$ is the minimizer of $h$ over $\mathcal{X}$. Then for $\hat{x}$ with*

$$\|\hat{x} - x_h\|_{\mathbf{M}} \leq \Delta, \ \|\nabla h(\hat{x})\|_{\mathbf{M}^\dagger} \leq G,$$

*and $\nabla h(\hat{x}), \nabla h(x_h) \in \text{Im}(\mathbf{M})$, we have for all $x \in \mathcal{X}$, $\langle \nabla h(\hat{x}), \hat{x} - x \rangle \leq 2L\Delta r + G\Delta$.*

*Proof.* First-order optimality of $x_h$ against $x \in \mathcal{X}$ implies $\langle \nabla h(x_h), x_h - x \rangle \leq 0$. The conclusion follows:

$$\langle \nabla h(\hat{x}), \hat{x} - x \rangle = \langle \nabla h(x_h), x_h - x \rangle + \langle \nabla h(\hat{x}) - \nabla h(x_h), x_h - x \rangle + \langle \nabla h(\hat{x}), \hat{x} - x_h \rangle$$

$$\leq 0 + 2L\Delta r + G\Delta.$$

$\square$

Putting together Lemma 35 and Lemma 36 we have the following corollary.

**Corollary 37.** *Consider a single iteration of Algorithm 7 from a pair of points $x_k, z_k$. Also, assume that $\|\bar{x} - x^*\|_{\mathbf{M}} \leq D$, where $x^*$ is the global optimizer of $f$. Then,*

$$f(y_k) + \langle \nabla f(y_k), z_{k+1} - y_k \rangle + \frac{\alpha}{2} \|y_k - z_{k+1}\|_{\mathbf{H}}^2 + \frac{1-\alpha}{2} \|z_k - z_{k+1}\|_{\mathbf{H}}^2$$

$$\leq f(x_{\bar{x},r}) + \frac{1-\alpha}{2} \|z_k - x_{\bar{x},r}\|_{\mathbf{H}}^2 - \frac{1}{2} \|z_{k+1} - x_{\bar{x},r}\|_{\mathbf{H}}^2 + L\Delta(5r + D).$$

*Proof.* First, the Hessian of the objective being minimized in (24) is $\mathbf{H}$, so the objective is $L$-smooth w.r.t $\|\cdot\|_{\mathbf{M}}$ over a region $\mathcal{B}_r(\bar{x})$ of bounded diameter $2r$. From Lemma 35 and 36 we have that the first-order optimality condition is correct up to an additive $2L\Delta r + G\Delta$, where $G$ is a bound on the gradient norm of the objective at $z_{k+1}$. The conclusion follows from $\mathbf{H}\mathbf{M}^{\dagger}\mathbf{H} \preceq L^2\mathbf{M}$ by smoothness, so that

$$G = \|\nabla f(y_k) + (1-\alpha)\mathbf{H}(z_{k+1} - z_k) + \alpha\mathbf{H}(z_{k+1} - y_k)\|_{\mathbf{M}^{\dagger}}$$
$$\leq \|\nabla f(y_k)\|_{\mathbf{M}^{\dagger}} + 2(1-\alpha)Lr + 2\alpha Lr \leq L(D+r) + 2Lr.$$

In the final inequality, we used $\|y_k - x^*\|_{\mathbf{M}} \leq \|\bar{x} - x^*\|_{\mathbf{M}} + \|y_k - \bar{x}\|_{\mathbf{M}} \leq D + r$, and Lipschitzness of $\nabla f$ w.r.t $\|\cdot\|_{\mathbf{M}}$. $\qquad\square$

With this in hand, we can quantify how much progress is made in each iteration of the algorithm.

**Lemma 38.** *Consider a single iteration of Algorithm 7 from a pair of points $x_k, z_k$. Also, assume that $\|\bar{x} - x^*\|_{\mathbf{M}} \leq D$, where $x^*$ is the global optimizer of $f$. Then,*

$$f(x_{k+1}) - f(x_{\bar{x},r}) + \frac{1}{2c}\|z_{k+1} - x_{\bar{x},r}\|_{\mathbf{H}}^2$$
$$\leq \left(1 - \frac{1}{c}\right)\left(f(x_k) - f(x_{\bar{x},r}) + \frac{1}{2c}\|z_k - x_{\bar{x},r}\|_{\mathbf{H}}^2\right) + Lc^{-1}\Delta(5r + D).$$

*Proof.* By stability and $x_{k+1} - y_k = \alpha(z_{k+1} - y_k) + (1-\alpha)(x_k - y_k)$ from the definition of the algorithm,

$$f(x_{k+1}) \leq f(y_k) + \langle\nabla f(y_k), x_{k+1} - y_k\rangle + \frac{c}{2}\|x_{k+1} - y_k\|_{\mathbf{H}}^2$$
$$= (1-\alpha)\left(f(y_k) + \langle\nabla f(y_k), x_k - y_k\rangle\right) + \alpha\left(f(y_k) + \langle\nabla f(y_k), z_{k+1} - y_k\rangle\right)$$
$$+ \frac{c}{2}\|\alpha(z_{k+1} - y_k) + (1-\alpha)(x_k - y_k)\|_{\mathbf{H}}^2$$
$$\leq (1-\alpha)f(x_k) + \alpha\left(f(y_k) + \langle\nabla f(y_k), z_{k+1} - y_k\rangle + \frac{1}{2}\|z_{k+1} - (1-\alpha)z_k - \alpha y_k\|_{\mathbf{H}}^2\right)$$
$$\leq (1-\alpha)f(x_k)$$
$$+ \alpha\left(f(y_k) + \langle\nabla f(y_k), z_{k+1} - y_k\rangle + \frac{\alpha}{2}\|y_k - z_{k+1}\|_{\mathbf{H}}^2 + \frac{1-\alpha}{2}\|z_k - z_{k+1}\|_{\mathbf{H}}^2\right).$$

The second inequality used convexity and $(1-\alpha)(\alpha z_k + x_k) = (1-\alpha^2)y_k$, which implies

$$\alpha(z_{k+1} - y_k) + (1-\alpha)(x_k - y_k) = \alpha(z_{k+1} - (1-\alpha)z_k - \alpha y_k),$$

and the third inequality used convexity of the norm squared. Substituting the earlier bound from Corollary 37 yields the conclusion, recalling $\alpha = c^{-1}$. $\qquad\square$

Now we are ready to prove the main result for the implementation of the ball optimization oracle, restated below.

**Theorem 8.** *Let $f$ be $L$-smooth, $\mu$-strongly convex, and $(r, c)$-Hessian stable in the seminorm $\|\cdot\|_{\mathbf{M}}$. Then, Algorithm 7 (in Appendix D.2) implements a $(\delta, r)$-ball optimization oracle for query point $\bar{x}$ with $\|\bar{x} - x^*\|_{\mathbf{M}} \leq D$ for $x^*$ the minimizer of $f$, and requires*

$$O\left(c\log^2\left(\frac{\kappa(D+r)c}{\delta}\right)\right)$$

*linear system solves in matrices of the form $\mathbf{H} + \lambda\mathbf{M}$ for nonnegative $\lambda$, where $\kappa = L/\mu$.*

*Proof.* First, for each iteration $k$, define the potential function

$$\Phi_k \stackrel{\text{def}}{=} f(x_k) - f(x_{\bar{x},r}) + \frac{1}{2c}\|z_k - x_{\bar{x},r}\|_{\mathbf{H}}^2.$$

By applying Lemma 38, and defining $E \stackrel{\text{def}}{=} L\Delta(5r + D) = \frac{\mu\delta^2}{4c}$ by the definition of $\Delta$ in Algorithm 7, we have

$$\Phi_{k+1} \leq \left(1 - \frac{1}{c}\right)\Phi_k + \frac{E}{c}.$$

Telescoping this guarantee and bounding the resulting geometric series in $E$ yields

$$\Phi_k \leq \left(1 - \frac{1}{c}\right)^k \Phi_0 + E. \tag{25}$$

Now, recalling $x_0 = z_0 = \bar{x}$, we can bound the initial potential by

$$\Phi_0 \leq \langle \nabla f(x_{\bar{x},r}), \bar{x} - x_{\bar{x},r}\rangle + \frac{c}{2}\|\bar{x} - x_{\bar{x},r}\|_{\mathbf{H}}^2 + \frac{1}{2c}\|\bar{x} - x_{\bar{x},r}\|_{\mathbf{H}}^2 \leq LDr + Lcr^2.$$

where we used $\mathbf{H} \preceq L\mathbf{M}$. Next, note that whenever we have $\Phi_k \leq \mu\delta^2/2c$, we have

$$\frac{1}{2c}\|z_k - x_{\bar{x},r}\|_{\mathbf{H}}^2 \leq \frac{\mu\delta^2}{2c} \Rightarrow \|z_k - x_{\bar{x},r}\|_{\mathbf{M}} \leq \delta,$$

where we used $\mathbf{H} \succeq \mu\mathbf{M}$. Thus, as $E = \mu\delta^2/4c$, running for

$$k = O\left(c\log\left(\frac{Lc(Dr + cr^2)}{\mu\delta^2}\right)\right) = O\left(c\log\left(\frac{\kappa(D + r)c}{\delta}\right)\right) \tag{26}$$

iterations suffices to guarantee $\Phi_k \leq \mu\delta^2/2c$ via (25), and therefore implements a $(\delta, r)$-ball optimization oracle at $\bar{x}$. It remains to bound the complexity of each iteration. For this, we apply Proposition 34 with the parameter $\Delta = \mu\delta^2/(4Lc(5r + D))$, and compute

$$\|\mathbf{H}(\bar{x} - (1 - \alpha)z_k - \alpha y_k) + \nabla f(y_k)\|_{\mathbf{M}^\dagger} \leq L\|\bar{x} - (1 - \alpha)z_k - \alpha y_k\|_{\mathbf{M}} + \|\nabla f(y_k)\|_{\mathbf{M}^\dagger} \leq 2Lr + LD.$$

Altogether, the number of linear system solves in the step is then bounded by

$$O\left(\log\left(\frac{L^2(D + r)^2}{\mu^2} \cdot \frac{Lc(D + r)}{\mu\delta^2 r}\right)\right) = O\left(\log\left(\frac{\kappa(D + r)c}{\delta}\right)\right),$$

where the first term is due to the squared norm and $\mu^{-2}$, and the second is due to $(r\Delta)^{-1}$, in the bound of Proposition 34. The final bound follows from the assumption $\delta < r$. Combining with (26) yields the claim. $\qquad\square$

## E   Proof of Lemma 11

Here, we prove Lemma 11, which shows quasi-self-concordance implies Hessian stability.

**Lemma 11.** *If thrice-differentiable $f : \mathbb{R}^d \to \mathbb{R}$ is $M$-quasi-self-concordant with respect to norm $\|\cdot\|$, then it is $(r, \exp(Mr))$-Hessian stable with respect to $\|\cdot\|$.*

*Proof.* Let $x, y \in \mathbb{R}^d$ be arbitrary and let $x_t \overset{\text{def}}{=} x + t(y - x)$ for all $t \in [0, 1]$. Then for all $u \in \mathbb{R}^d$,

$$\frac{d}{dt}\left(\|u\|_{\nabla^2 f(x_t)}^2\right) = \frac{d}{dt}\left(u^\top \nabla^2 f(x_t)u\right) = \nabla^3 f(x_t)[u, u, y - x].$$

The result follows from

$$\left|\log\left(\|u\|_{\nabla^2 f(y)}^2\right) - \log\left(\|u\|_{\nabla^2 f(x)}^2\right)\right| = \left|\int_0^1 \frac{\nabla^3 f(x_t)[u, u, y - x]}{\|u\|_{\nabla^2 f(x_t)}^2}dt\right| \leq M\|y - x\|.$$

$\qquad\square$

## F   Proofs for applications

**Corollary 12.** *Let $f(x) = g(\mathbf{A}x)$, for $g : \mathbb{R}^n \to \mathbb{R}$ that is $L$-smooth, $M$-QSC in the $\ell_2$ norm, and $\mathbf{A} \in \mathbb{R}^{n \times d}$. Let $x^*$ be a minimizer of $f$, and suppose that $\|x_0 - x^*\|_{\mathbf{M}} \leq R$ and $f(x_0) - f(x^*) \leq \epsilon_0$ for some $x_0 \in \mathbb{R}^d$, where $\mathbf{M} \overset{\text{def}}{=} \mathbf{A}^\top \mathbf{A}$. Then, Algorithm 3 yields an $\epsilon$-approximate minimizer to $f$ in*

$$O\left((RM)^{2/3}\log\left(\frac{\epsilon_0}{\epsilon}\right)\log^3\left(\frac{LR^2}{\epsilon}(1 + RM)\right)\right)$$

*linear system solves in matrices of the form $\mathbf{A}^\top\left(\nabla^2 g(\mathbf{A}x) + \lambda\mathbf{I}\right)\mathbf{A}$ for $\lambda > 0$ and $x \in \mathbb{R}^d$.*

*Proof.* Let the minimizer of $\tilde{f}(x)$ be $\tilde{x}$: observe by Lemma 41 that $\|x_0 - \tilde{x}\|_{\mathbf{M}} \leq \|x_0 - x^*\|_{\mathbf{M}} \leq R$. Note that $\tilde{f}(x)$ is $L + \epsilon/55R^2$-smooth and $\epsilon/55R^2$-strongly convex in $\|\cdot\|_{\mathbf{M}}$, and since the iterates of Algorithm 1 never are more than $D = 3\sqrt{2}R$ away from $\tilde{x}$ (Lemma 27), by the triangle inequality and $(1 + 3\sqrt{2})^2 \leq 55/2$, $\tilde{f}$ approximates $f$ to an additive error $\epsilon/2$ for all iterates. Next, letting $r = 1/M$, it follows from Lemma 11 that $g$ is $(r, e)$-Hessian stable in $\ell_2$, so that $f$ is $(r, e)$-Hessian stable in $\|\cdot\|_{\mathbf{M}}$ (see Lemma 39). It follows from the definition of Hessian stability that $\tilde{f}$ is also $(r, e)$-Hessian stable in $\|\cdot\|_{\mathbf{M}}$ (see Lemma 40). Finally, the conclusion follows from combining the guarantees of Theorem 6 and Theorem 8, where it suffices to minimize $\tilde{f}$ to $\epsilon/2$ additive error. $\square$

**Lemma 39.** *Let $g : \mathbb{R}^n \to \mathbb{R}$ be $M$-QSC in $\ell_2$. Then, $f(x) = g(\mathbf{A}x)$ is $M$-QSC in $\|\cdot\|_{\mathbf{A}^\top \mathbf{A}}$, for $\mathbf{A} \in \mathbb{R}^{n \times d}$, $f : \mathbb{R}^d \to \mathbb{R}$.*

*Proof.* Recall the condition on $g$ implies for all $u, h, x \in \mathbb{R}^d$,
$$\left|\nabla^3 g(\mathbf{A}x)[\mathbf{A}u, \mathbf{A}u, \mathbf{A}h]\right| \leq M\|\mathbf{A}h\|_2 \|\mathbf{A}u\|_{\nabla^2 g(y)}^2.$$
Using this, and recalling $\|\mathbf{A}h\|_2 = \|h\|_{\mathbf{A}^\top \mathbf{A}}$, $\nabla^2 f(x) = \mathbf{A}^\top \nabla^2 g(\mathbf{A}x)\mathbf{A}$, the result follows:
$$\left|\nabla^3 f(x)[u, u, h]\right| = \left|\nabla^3 g(\mathbf{A}x)[\mathbf{A}u, \mathbf{A}u, \mathbf{A}h]\right| \leq M\|h\|_{\mathbf{A}^\top \mathbf{A}} \|u\|_{\nabla^2 f(x)}^2.$$
$\square$

**Lemma 40.** *Suppose $f$ is $(r, c)$-stable in $\|\cdot\|$. Then for any matrix $\mathbf{M}$ and $\lambda \geq 0$, $\tilde{f}$ defined by $\tilde{f}(x) = f(x) + \frac{\lambda}{2}x^\top \mathbf{M}x$ is also $(r, c)$-stable in $\|\cdot\|$.*

*Proof.* It suffices to show that for $x, y \in \mathbb{R}^d$ with $\|x - y\| \leq r$,
$$c^{-1}\nabla^2 \tilde{f}(y) \preceq \nabla^2 \tilde{f}(x) \preceq c\nabla^2 \tilde{f}(y).$$
This immediately follows from $\nabla^2 \tilde{f}(x) = \nabla^2 f(x) + \lambda \mathbf{M}$, and combining
$$c^{-1}\nabla^2 f(y) \preceq \nabla^2 f(x) \preceq c\nabla^2 f(y), \ c^{-1}\lambda \mathbf{M} \preceq \lambda \mathbf{M} \preceq c\lambda \mathbf{M}.$$
$\square$

**Lemma 41.** *Let $f$ be a convex function with minimizer $x^*$, and let $\mathbf{M}$ be a positive semidefinite matrix. If $f_t(x) = f(x) + \frac{t}{2}\|x - y\|_{\mathbf{M}}^2$ is minimized at $x_t$, then for all $u \geq 0$, $\|x_u - y\|_{\mathbf{M}} \leq \|x^* - y\|_{\mathbf{M}}$.*

*Proof.* By the KKT conditions for $f_t$ we observe
$$\nabla f(x_t) = -t\mathbf{M}(x_t - y).$$
Taking derivatives of this with respect to $t$ we obtain
$$\nabla^2 f(x_t)\frac{dx_t}{dt} = -\mathbf{M}(x_t - y) - t\mathbf{M}\frac{dx_t}{dt}$$
or
$$\frac{dx_t}{dt} = -\left(\nabla^2 f(x_t) + t\mathbf{M}\right)^\dagger \mathbf{M}(x_t - y).$$
Now we have
$$\|x_u - y\|_{\mathbf{M}}^2 - \|x^* - y\|_{\mathbf{M}}^2 = 2\int_0^u \left(\frac{d}{dt}x_t\right)^\top \mathbf{M}(x_t - y)dt$$
$$= -2\int_0^u \|x_t - y\|_{\mathbf{M}(\nabla^2 f(x_t) + t\mathbf{M})^\dagger \mathbf{M}}^2 dt \leq 0$$
as desired. $\square$

**Lemma 42** (Approximation of $\mathrm{lse}_t$). *For all $y \in \mathbb{R}^n$,*
$$\left|\mathrm{lse}_t(y) - \max_{i \in [n]} y_i\right| < t \log n.$$

*Proof.* This follows from the facts that for $z \in \Delta^n$ the probability simplex, the entropy function $h(z) \stackrel{\text{def}}{=} \sum_{i \in [n]} z_i \log z_i$ has range $[-\log n, 0]$, $\max_{i \in [n]} y_i = \max_{z \in \Delta^n} z^\top y$, and by computation
$$\mathrm{lse}_t(y) = \max_{z \in \Delta^n} z^\top y - th(z).$$
$\square$

### F.1 Softmax calculus

*Proof of Lemma 14.* We will prove 1-smoothness and 2-QSC for lse, which implies the claims by chain rule. Let $S \overset{\text{def}}{=} \sum_{i \in [n]} \exp(x_i)$, and let $g \in \mathbb{R}^n$ with $g_i = \exp(x_i)/S$, $\mathbf{G} \overset{\text{def}}{=} \mathbf{diag}(g)$. Direct calculation reveals that for all $i, j, k \in [n]$

$$\frac{\partial}{\partial x_i} \mathrm{lse}(x) = g_i,$$

$$\frac{\partial^2}{\partial x_i \partial x_j} \mathrm{lse}(x) = \vec{1}_{i=j} g_i - g_i g_j, \text{ and}$$

$$\frac{\partial^3}{\partial x_i \partial x_j \partial x_k} \mathrm{lse}(x) = \vec{1}_{i=j=k} g_i - \vec{1}_{i=j} g_i g_j - \vec{1}_{i=k} g_i g_k - \vec{1}_{j=k} g_j g_k + 2 g_i g_j g_k.$$

Therefore, we have that $\nabla^2 \mathrm{lse}(x) = \mathbf{G} - g g^\top$. Now, note that $g_i \geq 0$ for all $i \in [n]$ and $\|g\|_1 = 1$. By Cauchy-Schwarz,

$$\left(g^\top u\right)^2 = \left(\sum_{i \in [n]} g_i u_i\right)^2 \leq \left(\sum_{i \in [n]} g_i\right)\left(\sum_{i \in [n]} g_i u_i^2\right) = u^\top \mathbf{G} u \leq \|g\|_1 \|u\|_\infty^2 = \|u\|_\infty^2.$$

This implies that $0 \preceq \nabla^2 \mathrm{lse}(x) \preceq \mathbf{G}$, and the first part follows. Further, letting $\mathbf{H} \overset{\text{def}}{=} \mathbf{diag}(h)$ and $\mathbf{U} \overset{\text{def}}{=} \mathbf{diag}(u)$ we have from direct calculation

$$u^\top \mathbf{G} \mathbf{U} h - (g^\top u)(h^\top \mathbf{G} u) = u^\top \nabla^2 \mathrm{lse}(x) \mathbf{U} h,$$

$$-(g^\top u)(h^\top \mathbf{G} u) + (g^\top u)^2 (g^\top h) = -(g^\top u)\left(u^\top \nabla^2 \mathrm{lse}(x) h\right)$$

$$-(u^\top \mathbf{G} u)(g^\top h) + (g^\top u)^2 (g^\top h) = -\left(u^\top \nabla^2 \mathrm{lse}(x) u\right)(g^\top h).$$

Combining these equations and the previous derivation of $\nabla^3 f(x)$,

$$\left|\nabla^3 \mathrm{lse}(x)[u, u, h]\right| = \left|u^\top \mathbf{G} \mathbf{H} u - 2(g^\top u)(h^\top \mathbf{G} u) - (u^\top \mathbf{G} u)(g^\top h) + 2(g^\top u)^2 (g^\top h)\right|$$

$$= \left|u^\top \nabla^2 \mathrm{lse}(x) \mathbf{U} h - (g^\top u)\left(u^\top \nabla^2 \mathrm{lse}(x) h\right) - \left(u^\top \nabla^2 \mathrm{lse}(x) u\right)(g^\top h)\right| \quad (27)$$

$$\leq \left|u^\top \nabla^2 \mathrm{lse}(x)\left(\mathbf{U} h - (g^\top u) h\right)\right| + \left|g^\top h\right| \|u\|_{\nabla^2 \mathrm{lse}(x)}^2.$$

Now, since $\nabla^2 \mathrm{lse}(x) \succeq 0$ we have

$$\left|u^\top \nabla^2 \mathrm{lse}(x)\left(\mathbf{U} h - (g^\top u) h\right)\right| \leq \|u\|_{\nabla^2 \mathrm{lse}(x)} \left\|\left(\mathbf{U} - (g^\top u)\mathbf{I}\right) h\right\|_{\nabla^2 \mathrm{lse}(x)}. \quad (28)$$

Further, recall $\nabla^2 \mathrm{lse}(x) \preceq \mathbf{G}$ and consequently

$$\left\|\left(\mathbf{U} - (g^\top u)\mathbf{I}\right) h\right\|_{\nabla^2 \mathrm{lse}(x)}^2 \leq \left\|\left(\mathbf{U} - (g^\top u)\mathbf{I}\right) h\right\|_{\mathbf{G}}^2 = \sum_{i \in [n]} h_i^2 g_i \left(u_i - g^\top u\right)^2$$

$$\leq \|h\|_\infty^2 \sum_{i \in [n]} g_i \left(u_i^2 - 2 u_i (g^\top u) + (g^\top u)^2\right) \quad (29)$$

$$= \|h\|_\infty^2 \left(\left(\sum_{i \in [n]} g_i u_i^2\right) - 2(g^\top u)^2 + \left(g^\top u\right)^2 \|g\|_1\right) \quad (30)$$

$$= \|h\|_\infty^2 \|u\|_{\nabla^2 \mathrm{lse}(x)}^2.$$

Combining (27), (28), (29), and using $|g^\top h| \leq \|g\|_1 \|h\|_\infty \leq \|h\|_\infty$ and $\|h\|_\infty \leq \|h\|_2$, the result follows. $\qquad \square$

### F.2 Proofs for Section 4.2

We first show a lemma that dicusses the linear system solve in Algorithm 3 used in solving the $\ell_\infty$ regression problem, which helps prove the main result as stated in Corollary 15.

**Lemma 43.** *Let $\hat{\mathbf{A}}$ be the vertical concatenation of $\mathbf{A}$ and $-\mathbf{A}$, and let $\mathbf{H}$ be a Hessian of the* lse *function. To solve a linear system in the form $\hat{\mathbf{A}}^\top (\mathbf{H} + \lambda \mathbf{I}) \hat{\mathbf{A}} x = \hat{\mathbf{A}}^\top b$ for $\lambda > 0$, it suffices to solve $O(1)$ linear systems of the form $\hat{\mathbf{A}}^\top \mathbf{D} \hat{\mathbf{A}} v = c$ for some positive-definite diagonal $\mathbf{D}$.*

*Proof.* Recall that the structure of the Hessian of softmax lets us write $\mathbf{H} + \lambda\mathbf{I} = \mathbf{D} - gg^\top$, for some diagonal $\mathbf{D} = \mathbf{G} + \lambda\mathbf{I}$, where $\mathbf{G} = \mathrm{diag}(g)$, and $g \geq 0$ entrywise has $\|g\|_1 = 1$. One can verify a solution of the linear system is

$$x = (\hat{\mathbf{A}}^\top\mathbf{D}\hat{\mathbf{A}})^\dagger\hat{\mathbf{A}}^\top b + \frac{1}{1 - g^\top\hat{\mathbf{A}}(\hat{\mathbf{A}}^\top\mathbf{D}\hat{\mathbf{A}})^\dagger\hat{\mathbf{A}}^\top g}\left((\hat{\mathbf{A}}^\top\mathbf{D}\hat{\mathbf{A}})^\dagger\hat{\mathbf{A}}^\top gg^\top\hat{\mathbf{A}}(\hat{\mathbf{A}}^\top\mathbf{D}\hat{\mathbf{A}})^\dagger\right)\hat{\mathbf{A}}^\top b,$$

where we use $\mathbf{U}^\dagger$ to denote the Moore-Penrose psuedo-inverse of $\mathbf{U}$. To show this is a valid formula, we also need to further verify that $g^\top\hat{\mathbf{A}}(\hat{\mathbf{A}}^\top\mathbf{D}\hat{\mathbf{A}})^\dagger\hat{\mathbf{A}}^\top g < 1$. This follows from

$$g^\top\hat{\mathbf{A}}(\hat{\mathbf{A}}^\top\mathbf{D}\hat{\mathbf{A}})^\dagger\hat{\mathbf{A}}^\top g = \langle(\hat{\mathbf{A}}^\top\mathbf{D}\hat{\mathbf{A}})^\dagger, \hat{\mathbf{A}}^\top gg^\top\hat{\mathbf{A}}\rangle < \langle(\hat{\mathbf{A}}^\top\mathbf{G}\hat{\mathbf{A}})^\dagger, \hat{\mathbf{A}}^\top\mathbf{G}\hat{\mathbf{A}}\rangle \leq 1.$$

It is thus straightforward from this formula that $x$ can be computed explicitly through a constant number of linear system solves in the form $\hat{\mathbf{A}}^\top\mathbf{D}\hat{\mathbf{A}}$ for some positive-definite $\mathbf{D} \succ 0$. $\qquad\square$

*Proof of Corollary 15.* This is a direct consequence of Corollary 12, where we note that the reduction in Lemma 43 applies, because of the form of the Hessian of $g$. $\qquad\square$

### F.2.1 Equivalent forms of $\ell_\infty$ regression

We first show the equivalence between the two formulations, i.e. $\min_{x\in\mathbb{R}^d}\|\mathbf{A}x - b\|_\infty$ and $\min_{y:\mathbf{A}^\top y=c}\|y\|_\infty$. Note this is the $\ell_\infty$ regression formulation used in our paper and in Ene and Vladu [18] respectively. To do this, for given $\mathbf{A} \in \mathbb{R}^{n\times d}$ we define the matrix

$$\mathbf{P}_\perp = \mathbf{I} - \mathbf{A}(\mathbf{A}^\top\mathbf{A})^\dagger\mathbf{A}^\top$$

to be the orthogonal projection to the complement of the column space of $\mathbf{A}$, so $\mathbf{A}^\top\mathbf{P}_\perp = \mathbf{P}_\perp\mathbf{A} = 0$.

For one side, it holds that

$$\min_{y:\mathbf{A}^\top y=c}\|y\|_\infty \quad\Longleftrightarrow\quad \min_{x\in\mathbb{R}^d}\|(\mathbf{A}^\top\mathbf{A})^\dagger\mathbf{A}^\top c + \mathbf{P}_\perp x\|_\infty.$$

This is because one can parametrize the space of $\{y|\mathbf{A}^\top y = c\}$ by $\{(\mathbf{A}^\top\mathbf{A})^\dagger\mathbf{A}^\top c + \mathbf{P}_\perp x|x \in \mathbb{R}^d\}$ based on the orthogonal decomposition of $y$ onto the column space of $\mathbf{A}$ and its complement. Thus, the constraint must hold and the objective function can be written equivalently as on the right hand side. For the other side, it holds that

$$\min_{x\in\mathbb{R}^d}\|\mathbf{A}x - b\|_\infty \quad\Longleftrightarrow\quad \min_{y:\mathbf{P}_\perp y=-\mathbf{P}_\perp b}\|y\|_\infty$$

by noticing the fact that $y = \mathbf{A}x - b$ for some $x \in \mathbb{R}^d$ if and only if $\mathbf{P}_\perp y = -\mathbf{P}_\perp b$, and then rewriting the constraints and objective function in terms of $y$ respectively.

In particular, our method applied to $\min_{x\in\mathbb{R}^d}\|(\mathbf{A}^\top\mathbf{A})^\dagger\mathbf{A}^\top c + \mathbf{P}_\perp x\|_\infty$ gives an alternative way to solve the problem considered in [18]. We may therefore use Corollary 15 directly, where it suffices to take the norm in the bound on domain size $R$ to be Euclidean because the projection matrix $\mathbf{P}_\perp$ is bounded by $\mathbf{I}$. This gives the following claim as in Corollary 44.

**Corollary 44.** *Let $\mathbf{A} \in \mathbb{R}^{n\times d}$, $c \in \mathbb{R}^d$, and $\epsilon > 0$. Suppose that the minimizer $y^*$ of the optimization problem $\min_{\mathbf{A}^\top y=c}\|y\|_\infty$ satisfies $\|y^* - y_0\|_2 \leq R$ for an initial point $y_0$. We can find $y$ satisfying $\mathbf{A}^\top y = c$ and $\|y\|_\infty \leq \|y^*\|_\infty + \epsilon$ using*

$$O\left(\left(\frac{R\log n}{\epsilon}\right)^{2/3}\log^4(nR/\epsilon)\right)$$

*linear system solves in matrices of the form $\mathbf{A}^\top\mathbf{D}\mathbf{A}$ for some diagonal $\mathbf{D} \succ 0$.*

To complete the proof of Corollary 44, the only thing remaining to show is that solving systems induced by matrices of the form $\mathbf{P}_\perp\mathbf{D}\mathbf{P}_\perp$ (as Corollary 15 requires) can be reduced to solving systems in matrices of the form $\mathbf{A}\mathbf{D}'\mathbf{A}^\top$ and $\mathbf{D}$, for some positive definite diagonal $\mathbf{D}'$. This is given formally through the following lemma.

**Lemma 45.** *For projection matrix* $\mathbf{P}_\perp = \mathbf{I} - \mathbf{A}(\mathbf{A}^\top\mathbf{A})^\dagger\mathbf{A}^\top$, *let* $\mathbf{D}$ *be a diagonal matrix and let* $g$ *be a vector where* $\mathbf{P}_\perp\mathbf{D}\mathbf{P}_\perp x = \mathbf{P}_\perp g$ *has a solution. Then*

$$x = \mathbf{D}^{-1}\left[g - \mathbf{A}(\mathbf{A}^\top\mathbf{D}^{-1}\mathbf{A})^\dagger\mathbf{A}^\top\mathbf{D}^{-1}g\right]$$

*is a solution to the linear system.*

*Proof.* By directly expanding terms, we have

$$
\begin{aligned}
\mathbf{P}_\perp\mathbf{D}\mathbf{P}_\perp x &= \mathbf{P}_\perp\mathbf{D}\left(\mathbf{I} - \mathbf{A}(\mathbf{A}^\top\mathbf{A})^\dagger\mathbf{A}^\top\right)\mathbf{D}^{-1}\left[g - \mathbf{A}(\mathbf{A}^\top\mathbf{D}^{-1}\mathbf{A})^\dagger\mathbf{A}^\top\mathbf{D}^{-1}g\right] \\
&= \mathbf{P}_\perp\left[g - \mathbf{A}(\mathbf{A}^\top\mathbf{D}^{-1}\mathbf{A})^\dagger\mathbf{A}^\top\mathbf{D}^{-1}g\right] \\
&= \mathbf{P}_\perp g,
\end{aligned}
$$

where for the last equality we use the fact that $\mathbf{P}_\perp\mathbf{A} = 0$. $\qquad\square$

### F.3 Proofs for Section 4.3

We first introduce a more general result showing the result generically applies whenever the QSC-ness of $f(x) = g(\mathbf{A}x)$ is inherited from the QSC-ness of one-dimensional loss functions $\{g_i(y)\}_{i\in[n]}$.

**Lemma 46.** *Given a coordinate-separable objective function* $g(y)$ *such that each coordinate* $g_i(y)$ *is* $M$-*QSC w.r.t* $\|\cdot\|_2$, *then* $f(x) = g(\mathbf{A}x)$ *is* $(M\cdot\max\|a_i\|_2)$-*QSC w.r.t* $\|\cdot\|_2$.

*Proof of Lemma 46.* By the definition of QSC, we have

$$\left|\nabla^3 g_i(y)[u_i, u_i, h_i]\right| \le M|h_i|\|u_i\|_{\nabla^2 g_i(y)}^2, \forall i \in [n].$$

Setting $u_i = \langle a_i, u\rangle$, $h_i = \langle a_i, h\rangle$ and summing over $i \in [n]$ gives

$$
\begin{aligned}
|\nabla^3 f(x)[u, u, h]| \le \sum_{i\in[n]}\left|\nabla^3 g_i(y)[a_i^\top u, a_i^\top u, a_i^\top h]\right| &\le \sum_{i\in[n]} M\|a_i\|_2\|h\|_2\|\langle a_i, u\rangle\|_{\nabla^2 g_i(y)}^2 \\
&\le M\left(\max_{i\in[n]}\|a_i\|_2\right)\|h\|_2\|u\|_{\nabla^2 f(x)}^2,
\end{aligned}
$$

where for the first and last inequality we use chain rule and separability of $g$, and thus also $f$. $\quad\square$

*Proof of Lemma 16.* Lemma 16 follows immediately from Lemma 46 as the logistic objective has the desired property according to Section 4.1. $\qquad\square$

*Proof of Lemma 17.* Note that

$$|\nabla^3 f(x)[u, u, h]| \le \frac{2}{t}\|\mathbf{A}h\|_\infty\|u\|_{\nabla^2 f(x)}^2 \le \frac{2}{t}\max_{i\in[n]}\|a_i\|_2\|h\|_2\|u\|_{\nabla^2 f(x)}^2,$$

where the first inequality follows from Lemma 14 and the definition of QSC. $\qquad\square$

For a given $x$, let $y = \mathbf{A}x$, $S \stackrel{\text{def}}{=} \sum_{i\in[n]}\exp(y_i)$, $g \in \mathbb{R}^n$ with $g_i = \exp(x_i)/S$, and $\mathbf{G} \stackrel{\text{def}}{=} \mathbf{diag}(g)$, one has $\nabla^2(\mathrm{lse}_t(\mathbf{A}x) + \frac{\epsilon}{4R^2}\|x - x_0\|^2) = \frac{1}{t}\mathbf{A}^\top(\mathbf{G} - gg^\top)\mathbf{A} + \frac{\epsilon}{2R^2}\mathbf{I}$. We provide the following bound on the cost of solving a linear system in a Hessian of the objective in our procedure.

**Lemma 47** (First-order method for linear system solve). *Let* $\mathbf{U} \stackrel{\text{def}}{=} \frac{1}{t}\mathbf{A}^\top\mathbf{G}\mathbf{A} + \hat\lambda\mathbf{I}$ *where* $\hat\lambda \stackrel{\text{def}}{=} \frac{\epsilon}{2R^2} + \lambda$ *and* $v \stackrel{\text{def}}{=} \frac{1}{\sqrt{t}}\mathbf{A}^\top g$. *We can solve linear systems of the form*

$$\left[\mathbf{U} - vv^\top\right]x = b$$

*for vector* $b \in \mathrm{im}(\mathbf{A})$ *within runtime*

$$\widetilde{O}\left(nd + d^{1.5}\frac{\max_{i\in[n]}\|a_i\|_2 R}{\sqrt{\epsilon t}}\right).$$

*Proof.* By the Sherman-Morrison formula, we can solve

$$x = [\mathbf{U} - vv^\top]^{-1}b = \mathbf{U}^{-1}b + \frac{\mathbf{U}^{-1}vv^\top\mathbf{U}^{-1}}{1 - v^\top\mathbf{U}v}b,$$

which reduces the problem to solving to high precision linear systems of the form $\mathbf{U}x = \hat{b}$ for $\hat{b} = b$ and $\hat{b} = vv^\top\mathbf{U}^{-1}b$. It is straightforward to see $\hat{b} \in \mathrm{im}(\mathbf{A})$ and thus, letting $\hat{b} = \mathbf{A}^\top\hat{c}$, the problem is equivalent to solving the linear regression problem

$$\min_x \left\| \frac{1}{\sqrt{t}}\mathbf{G}^{1/2}\mathbf{A}x - \sqrt{t}\mathbf{G}^{-1/2}\hat{c} \right\|_2^2 + \frac{\hat{\lambda}}{2}\|x\|_2^2 = \min_x \left\| \hat{\mathbf{A}}x - b \right\|_2^2,$$

where we define $\hat{\mathbf{A}} \stackrel{\text{def}}{=} [1/\sqrt{t}\mathbf{G}^{1/2}\mathbf{A}; \sqrt{\hat{\lambda}/2} \cdot \mathbf{I}]$ and use $b$ to denote $[\sqrt{t}\mathbf{G}^{-1/2}\hat{c}; 0]$ in an abuse of notation. Using the accelerated regression solver of Agarwal et al. [5] (cf. Theorem 5), we can solve this regression problem in time

$$\widetilde{O}\left( nd + d^{1.5}\sqrt{\frac{\mathrm{tr}(\hat{\mathbf{A}}^\top\hat{\mathbf{A}})}{\lambda_{\min}(\hat{\mathbf{A}}^\top\hat{\mathbf{A}})}} \right). \tag{31}$$

We now bound these terms. By definition of $\hat{\mathbf{A}}$,

$$\hat{\mathbf{A}}^\top\hat{\mathbf{A}} = \frac{1}{t}\mathbf{A}^\top\mathbf{G}\mathbf{A} + \frac{\hat{\lambda}}{2}\mathbf{I} \succeq \frac{\hat{\lambda}}{2}\mathbf{I},$$

$$\mathrm{tr}\left( \hat{\mathbf{A}}^\top\hat{\mathbf{A}} \right) = \mathrm{tr}\left( \frac{1}{t}\mathbf{A}^\top\mathbf{G}\mathbf{A} \right) + \frac{\hat{\lambda}}{2}d = \frac{1}{t}\sum_{i\in[n]} g_i\|a_i\|^2 + \frac{\hat{\lambda}}{2}d \le \frac{1}{t}\max_i\|a_i\|^2 + \frac{\hat{\lambda}}{2}d,$$

where the last inequality follows from the fact that $\sum_{i\in[n]} g_i = 1$. Plugging these bounds back into the runtime (31) and combining with the Sherman-Morrison procedure gives an overall runtime of

$$\widetilde{O}\left( nd + d^{1.5}\frac{\max_{i\in[n]}\|a_i\|_2 R}{\sqrt{\epsilon t}} \right)$$

for solving the linear system (note that the bound (31) is worst when $\lambda = 0$). $\qquad\square$

This further implies the following claim.

**Lemma 48.** *Let* $h(x) = \mathrm{lse}_t(\mathbf{A}x) + \frac{\epsilon}{4R^2}\|x - x_0\|_2^2$ *for* $R = \|x_0 - x^*\|_2$ *and* $t = \frac{\epsilon}{2\log n}$. *Given a point* $z$ *we can solve linear systems of the form*

$$\left( \nabla^2 h(z) \right) x = b$$

*in time*

$$\widetilde{O}\left( nd + d^{1.5}\frac{\max_{i\in[n]}\|a_i\|_2 R}{\sqrt{\epsilon t}} \right).$$

We thus obtain the following result for using the first-order method in linear system solves in Algorithm 3 by combining the QSC condition in Lemma 17, and the efficient first-order method for each linear system solve in Lemma 48.

**Corollary 18.** *With initial function error* $\epsilon_0$ *and* $R = \|x_0 - x^*\|_2$, *Algorithm 3 using the first-order linear system solver of Agarwal et al. [5] returns an* $\epsilon$-*approximate minimizer within total runtime* $\widetilde{O}\left( \left(\max_{i\in[n]}\|a_i\|_2 \frac{R}{\epsilon}\right)^{2/3} \left(nd + d^{1.5}\max_{i\in[n]}\|a_i\|_2 \frac{R}{\epsilon}\right) \right)$.

## F.4  Proofs for $\ell_p$ regression

We refer to the optimal value of (7) by $f^*$, and its minimizer by $x^*$; we will solve (7) to $1 + \delta$ multiplicative accuracy. By taking $p$th roots and solving to an appropriate lower accuracy level, this also recovers more standard formulations of minimizing $\|\mathbf{A}x - b\|_p$.

Prior work on this problem shows (7) can be minimized using fewer than the $O(n^{1/2})$ linear system solves that an interior point method would require: the state of the art algorithms

of Adil and Sachdeva [1], Adil et al. [2] minimize $f$ to $1 + \delta$ multiplicative accuracy by solving $\widetilde{O}\left(\min\left(pn^{1/3}, p^{O(p)}n^{\frac{p-2}{3p-2}}\right)\log(1/\delta)\right)$ linear systems in $\mathbf{A}^\top \mathbf{D} \mathbf{A}$ where $\mathbf{D}$ is a positive semidefinite diagonal matrix. In this section we provide an algorithm to minimize $g$ in $\widetilde{O}(p^{14/3}n^{1/3}\log^4(n/\delta))$ such systems. While our techniques do not improve on the state of the art, we believe our proof and algorithm are simpler than the previous work and of independent interest.

Algorithm 8 summarizes our approach. It consists iteratively applying Algorithm 3 to the objective (7) with exponentially shrinking target additive error. We initialize the algorithm at $x_0 = \arg\min_x \|\mathbf{A}x - b\|_2$. Using the fact that $\|y\|_2 \le n^{(p-2)/2p}\|y\|_p$ for all $y$ and $p$, the initialization satisfies

$$\epsilon_0 \overset{\text{def}}{=} \|\mathbf{A}x_0 - b\|_p^p \le r\|\mathbf{A}x_0 - b\|_2^p \le \|\mathbf{A}x^* - b\|_2^p \le n^{(p-2)/2}f^*. \tag{32}$$

The algorithm maintains the invariant

$$f(x_k) - f^* \le (2^{-p})^k \epsilon_0 \le (2^{-k}n)^p,$$

so that running $k = \log_2 \frac{n}{\delta^{1/p}}$ iterations guarantees multiplicative error of at most $\delta^4$.

Unlike the previous two applications, the function $g$ is *not* QSC, as its Hessian is badly behaved near zero. Nevertheless we argue that an $\ell_2$ regularization of $g$ is QSC (Lemma 49), and—because Algorithm 3 includes such regularization—the conclusion of the corollary still holds (Lemma 52). The key to our analysis is showing that with each iteration the distance to the optimum $R$ shrinks (due to convergence to $x^*$) by the same factor that the QSC constant $M$ grows (due to diminishing regularization), such that $RM = O(p\sqrt{n})$ throughout, leading to the overall $\mathsf{poly}(p)n^{1/3}$ complexity guarantee.

---

**Algorithm 8** High accuracy $\ell_p$ regression

---

1: **Input:** $\mathbf{A} \in \mathbb{R}^{n \times d}$, $b \in \mathbb{R}^n$, multiplicative error tolerance $\delta \ge 0$.
2: Set $x_0 = \mathbf{A}^\dagger b$ and $\epsilon_0 = f(x_0) = \|\mathbf{A}x_0 - b\|_p^p$.
3: **for** $k \le \log_2(n/\delta^{1/p})$ **do**
4: $\quad \epsilon_k \leftarrow 2^{-p}\epsilon_{k-1}$
5: $\quad x_k \leftarrow$ output of Algorithm 3 applied on $f(x) = \|\mathbf{A}x - b\|_p^p$ with initialization $x_{k-1}$, desired accuracy $\epsilon_k$ and parameters $R = O(n^{(p-2)/2p}\epsilon_k^{1/p})$ and $M = O(p\sqrt{n}/R)$ (see Lemma 52)
6: **end for**

---

We first bound the QSC of $\ell_2$ regularization of $g$.

**Lemma 49.** *For any $b \in \mathbb{R}^n$, $y \in \mathbb{R}^d$, $p \ge 3$, $\mu \ge 0$, the function $g(x) + \mu\|x-y\|_2^2$ is $O(p\mu^{-1/(p-2)})$-QSC with respect to $\ell_2$.*

*Proof.* Let $\tilde{g}(x) = g(x) + \mu\|x-y\|_2^2$. We observe that

$$|\nabla^3 \tilde{g}(x)[h, u, u]| = p(p-1)(p-2)\sum_{i=1}^n h_i u_i^2 |x_i - b_i|^{p-3}$$

and

$$\nabla^2 \tilde{g}(x)[u, u] = \sum_{i=1}^n u_i^2 \left(p(p-1)|x - b|_i^{p-2} + 2\mu\right).$$

Now,

$$|\nabla^3 \tilde{g}(x)[h,u,u]| \leq p(p-1)(p-2)\|h\|_\infty \sum_{i=1}^n u_i^2 |x_i - b_i|^{p-3}$$

$$\leq \|h\|_2 \sum_{i=1}^n (p(p-1)(p-2))^{\frac{p}{3p-6}} u_i^2 \left((p(p-1)(p-2))^{\frac{2}{3}}|x_i - b_i|^{p-2}\right)^{\frac{p-3}{p-2}}$$

$$\leq O(p\mu^{-1/(p-2)})\|h\|_2 \sum_{i=1}^n u_i^2 \left(p(p-1)|x_i - b_i|^{p-2}\right)^{\frac{p-3}{p-2}} \mu^{\frac{1}{p-2}}$$

$$\leq O(p\mu^{-1/(p-2)})\|h\|_2 \sum_{i=1}^n u_i^2 \left(p(p-1)|x_i - b_i|^{p-2} + 2\mu\right)$$

$$\leq O(p\mu^{-1/(p-2)})\|h\|_2 \nabla^2 \tilde{g}(x)[u,u]$$

where we used that $u_i^2 |x_i - b_i|^{p-3}$ is nonnegative in the first line and $\|\cdot\|_\infty \leq \|\cdot\|_2$ in the second. In the third line we used that $(p(p-1)(p-2))^{\frac{p}{3p-6}} \leq p^{\frac{3p}{3p-6}} = p p^{\frac{6}{3p-6}} = O(p)$ since $p^{\frac{6}{3p-6}}$ is at most 9 if $p \geq 3$. Finally in the fourth line we applied the inequality $x^\alpha y^{1-\alpha} \leq \max(x,y) \leq x + y$ for nonnegative $x, y$, and $\alpha \in [0,1]$. The claim follows. $\qquad\square$

We next show approximate minimizers of $f$ are close to $x^*$.

**Lemma 50.** *For $x \in \mathbb{R}^d$ with $f(x) - f^* \leq \epsilon$, we have $\|x - x^*\|_M^p \leq 2^p n^{\frac{p-2}{2}}\epsilon$.*

To prove Lemma 50 we use the following lemma from [2], with notation modified to our setting.[5]

**Lemma 51** (Adil et al. [2, Lemma 4.5]). *Let $p \in (1,\infty)$. Then for any two vectors $y, \Delta \mathbb{R}^n$,*

$$\|y\|_p^p + v^\top \Delta + \frac{p-1}{p \cdot 2^p}\|\Delta\|_p^p \leq \|y + \Delta\|_p^p$$

*where $v_i = p|y_i|^{p-2}y_i$ is the gradient of $\|y\|_p^p$.*

*Proof.* Substituting $y = \mathbf{A}x^* - b$, $\Delta = \mathbf{A}(x - x^*)$ in Lemma 51, and simplifying gives

$$\|\mathbf{A}x^* - b\|_p^p + \nabla f(x^*)^\top (x - x^*) + \frac{p-1}{p2^p}\|\mathbf{A}(x - x^*)\|_p^p \leq \|\mathbf{A}x - b\|_p^p.$$

As $\nabla f(x^*)^\top (x - x^*) = 0$ by optimality of $x^*$, we obtain

$$\frac{p-1}{p2^p}\|\mathbf{A}(x - x^*)\|_p^p \leq \|\mathbf{A}x - b\|_p^p - \|\mathbf{A}x^* - b\|_p^p = f - f^* \leq \epsilon.$$

Now using $\|\cdot\|_2 \leq n^{\frac{p-2}{2p}}\|\cdot\|_p$ this implies

$$\|x - x^*\|_M^p \leq 2^{p+1} n^{\frac{p-2}{2}}\epsilon.$$

as $\frac{p}{p-1} \leq 2$ for $p \geq 3$. $\qquad\square$

Finally, we bound the complexity of executions of Line 5.

**Lemma 52.** *Let $\epsilon_{k-1} \geq \delta f^*$. Initialized at $x_{k-1}$ satisfying $f(x_{k-1}) - f^* \leq \epsilon_{k-1}$, Algorithm 3 computes $x_k$ with $f(x_k) - f^* \leq 2^{-p}\epsilon_{k-1} = \epsilon_k$ in $O(p^{14/3}n^{1/3}\log^3(n/\delta))$ linear system solves in $\mathbf{A}^\top \mathbf{D}\mathbf{A}$ for diagonal matrix $\mathbf{D} \succeq 0$.*

*Proof.* We apply Algorithm 3 to compute an $\epsilon_k = 2^{-p}\epsilon_{k-1}$ approximate minimizer of $f$ in

$$O\left((RM)^{2/3}\log^3\left(\frac{LR^2}{\epsilon_k}(1 + MR)\right)\log\left(\frac{\epsilon_{k-1}}{\epsilon_k}\right)\right)$$

linear system solutions, with parameters $R$, $M$ and $L$ that we bound as follows.

By Lemma 50,

$$\|x_{k-1} - x^*\|_{\mathbf{M}}^p \leq 2^{p+1} n^{\frac{p-2}{2}} \epsilon_{k-1} \stackrel{\text{def}}{=} R^p$$

We add $\frac{\epsilon_k}{55R^2}\|x - x_k\|_{\mathbf{M}}^2$ to $f$ in obtaining $\tilde{f}$, and observe that the proof of Corollary 12 only requires us to show that $\tilde{f}$ is QSC. By Lemma 49, we see that $\tilde{f}$ is $M = O\left(p\left(R^2/\epsilon_k\right)^{1/(p-2)}\right)$-QSC. Therefore, for any $p \geq 3$ we have

$$RM = O\left(pR^{\frac{p}{p-2}}\epsilon_k^{-\frac{1}{p-2}}\right) = O\left(p\sqrt{n}\left(\frac{2^{p+1}\epsilon_{k-1}}{\epsilon_k}\right)^{\frac{1}{p-2}}\right) = O(p\sqrt{n}),$$

so the polynomial term in the running time is at most $O\left((RM)^{2/3}\right) = O\left(p^{2/3}n^{1/3}\right)$. We now bound the logarithmic factors in the runtime. Observe that for any $x$ output by our MS oracle implementation we have that $\|x - x^*\|_{\mathbf{M}} \leq 2\sqrt{3}R$ (Lemma 27 with $\sigma = \frac{1}{2}$). As the Hessian of $f$ is $\mathbf{A}^\top \mathbf{D}\mathbf{A}$ where $\mathbf{D}_{ii} = p(p-1)|\mathbf{A}x - b|_i^{p-2}$ we may upper bound the smoothness of $f$ (w.r.t. $\|\cdot\|_{\mathbf{M}}$) at all points encountered during the algorithm by

$$O\left(p^2 \max_{\|x-x^*\|_{\mathbf{M}} \leq 2\sqrt{3}R} \|\mathbf{A}x - b\|_\infty^{p-2}\right).$$

For any $x$ such that $\|x - x^*\|_{\mathbf{M}} \leq 2\sqrt{3}R$ we have $\|\mathbf{A}x - b\|_\infty \leq \|\mathbf{A}x^* - b\|_\infty + \|\mathbf{A}(x - x^*)\|_\infty \leq \|\mathbf{A}x^* - b\|_p + \|x - x^*\|_{\mathbf{M}} \leq (f^*)^{1/p} + 2\sqrt{3}R$. Using the assumption $f^* \leq \epsilon_{k-1}/\delta \leq \delta^{-1}n^{-\frac{p-2}{2}}R^p$, we may upper bound $L$ as

$$L = O\left(4^p p^2 \left(1 + \delta^{-\frac{1}{p}}n^{-\frac{p-2}{2p}}\right)^{p-2} R^{p-2}\right).$$

Recalling that $R^p = 2^{2p+1}n^{\frac{p-2}{2}}\epsilon_k$, we obtain

$$\frac{LR^2}{\epsilon_k}(1 + MR) = O\left(4^p p^2 \left(1 + \delta^{-\frac{1}{p}}n^{-\frac{p-2}{2p}}\right)^{p-2} \frac{R^p}{\epsilon_k}(1 + p\sqrt{n})\right)$$

$$= O\left(16^p \left(\sqrt{n} + (n/\delta)^{1/p}\right)^{p-2} p^3\sqrt{n}\right) = O\left(17^p \left(\sqrt{n} + (n/\delta)^{1/p}\right)^p\right)$$

Taking a logarithm yields

$$\log\left(\frac{LR^2}{\epsilon}(1 + MR)\right) \leq O\left(p\log n + \log\frac{n}{\delta}\right) = O\left(p\log\frac{n}{\delta}\right).$$

Finally since $\log(\epsilon_{k-1}/\epsilon_k) = p$, combining the above bounds with the running time of Corollary 12 gives a bound of $O(p^{14/3}n^{1/3}\log^3(n/\delta))$ linear system solves as desired. $\qquad\square$

For proving Corollary 19, our final runtime follows from Lemma 52 and the fact that the loop in Algorithm 8 repeats $O(\log\frac{n}{\delta})$ times.

## G  Lower bound

We now provide a detailed derivation and discussion of our lower bound. For simplicity, we focus on a setting where the functions are defined on a bounded domain of radius $R > 0$, and are 1-Lipschitz but potentially non-smooth; afterwards, we explain how to extend the result to unconstrained, differentiable and strictly convex functions. We assume throughout the section that $\mathbf{M} = \mathbf{I}$, i.e., that we work in the standard $\ell_2$ norm.

Following the literature on information-based complexity [26], we state and prove our lower bound for the class of *r-local oracles*, which for every query point $\bar{x}$ return a *function* $f_{\bar{x}}$ that is identical to $f$ in a neighborhood of $x$. However, we additionally require the radius of this neighborhood to be at least $r$. Therefore, a query to an $r$-local oracle suffices to implement a ball optimization oracle (as well as a gradient oracle), and consequently a lower bound on algorithms interacting with an $r$-local oracles is also a lower bounds for algorithms a utilizing ball optimization oracle. The formal definition of the oracle class follows.

**Definition 53** (Local oracles and algorithms). We call $\mathcal{O}_{\text{local}}$ an $r$-local oracle for function $f$ : $\mathcal{B}_R(0) \to \mathbb{R}$ if given query point $\bar{x} \in \mathbb{R}^d$ it returns $f_{\bar{x}} : \mathcal{B}_R(0) \to \mathbb{R}$ such that $f_{\bar{x}}(x) = f(x)$ for all $x \in \mathcal{B}_r(\bar{x})$. We call (possibly randomized) algorithms that interact with $r$-local oracles $r$-*local algorithms*.

We prove our lower bound using a small extension of the well-established machinery of high-dimensional optimization lower bounds [26, 29, 12, 9]. To describe it, we start with the notion of coordinate progress, denoting for any $x \in \mathbb{R}^d$

$$i_r^+(x) \stackrel{\text{def}}{=} \min\{i \in [d] \mid |x_j| \le r \text{ for all } j \ge i\}, \tag{33}$$

where we let $i_r^+(x) \stackrel{\text{def}}{=} d + 1$ when $|x_d| > r$, i.e. $i_r^+(x)$ is the index following the last "large" entry of $x$. With this notation, we define a key notion for proving our lower bound.

**Definition 54** (Robust zero-chains). Function $f : B_1(0) \to \mathbb{R}$ is an $r$-robust zero-chain if $\forall \bar{x} \in \mathbb{R}^d$, $x \in \mathcal{B}_r(\bar{x})$,
$$f(x) = f(x_1, \dots, x_{i_r^+(\bar{x})}, 0, \dots, 0).$$

The notion of $r$-robust zero-chain we use here is very close to the robust zero-chain defined in [12, Definition 4], except here we require the equality to hold in a fixed ball rather than just a neighborhood of $\bar{x}$. The following lemma shows that $r$-local algorithms operating on a random rotation of an $r$-robust zero-chain make slow progress with high probability.

**Lemma 55.** *Let* $\frac{r}{R}, \delta \in (0,1)$, $N \in \mathbb{N}$ *and* $d \ge \left\lceil N + \frac{20R^2}{r^2} \log \frac{20NR^2}{\delta r^2} \right\rceil$. *Let* $f : \mathcal{B}_R(0) \to \mathbb{R}$ *be an $r$-robust zero-chain and let* $\mathbf{U} \in \mathbb{R}^{d \times d}$ *be a random orthogonal matrix and fix an $r$-local algorithm $\mathcal{A}$. With probability at least $1 - \delta$ over the draw of $\mathbf{U}$, there exists an $r$-local oracle $\mathcal{O}$ for* $f_{\mathbf{U}}(x) \stackrel{\text{def}}{=} f(\mathbf{U}^\top x)$ *such that the queries $x_1, x_2, \dots$ of $\mathcal{A}$ interacting with $\mathcal{O}$ satisfy*

$$i_r^+(\mathbf{U}^\top x_i) \le i \quad \text{for all } i \le N.$$

We provide a concise proof of Lemma 55 in Section G.1 below, where we also compare it to existing proofs in the literature.

With Lemma 55 in hand, to prove the lower bound we need to construct an $r$-robust zero-chain function $f_{N,r}$ with the additional property that every $x$ with $i_r^+(x) \le N$ is significantly suboptimal. Fortunately, Nemirovski's function [26] satisfies these properties.

**Lemma 56.** *Let $r > 0$ and $N \in \mathbb{N}$. Define*

$$f_{N,r}(x) \stackrel{\text{def}}{=} \max_{i \in [N]}\{x_i - 4r \cdot i\} \tag{34}$$

1. *The function $f_{N,r}$ is an $r$-robust zero-chain.*

2. *For all $x \in \mathcal{B}_R(0)$ such that $i_r^+(x) \le N$, we have $f_{N,r}(x) - \inf_{z \in \mathcal{B}_R(0)} f_{N,r}(z) \ge \frac{R}{\sqrt{N}} - 4Nr$.*

3. *The function $f_{N,r}$ is convex and 1-Lipschitz.*

*Proof.* To prove the first part, fix $\bar{x}$, $x \in \mathcal{B}_r(\bar{x})$ and $j > i_r^+(\bar{x})$. We have for all $x \in \mathcal{B}_r(\bar{x})$ that $|x_k - \bar{x}_k| \le r$ for all $k \in [d]$, and therefore

$$x_j - 4r \cdot j \stackrel{(i)}{\le} \bar{x}_j + r - 4r \cdot j \stackrel{(ii)}{\le} \bar{x}_{i_r^+(\bar{x})} + 3r - 4r \cdot j \stackrel{(iii)}{\le} x_{i_r^+(\bar{x})} + 4r - 4r \cdot j \stackrel{(iv)}{\le} x_{i_r^+(\bar{x})} - 4r \cdot i_r^+(\bar{x}).$$

Transitions $(i)$ and $(iii)$ above are due to $\|x - \bar{x}\| \le r$; transition $(ii)$ is due to the definition (33) of $i_r^+$, which implies $|\bar{x}_{i_r^+(\bar{x})}| \le r$ and $|\bar{x}_j| \le r$; and $(iv)$ is due to $j > i_r^+(\bar{x})$. Consequently, we have

$$f_{N,r}(x) = \max_{i \in [i_r^+(\bar{x})]}\{x_i - 4r \cdot i\} \quad \text{for all } x \in \mathcal{B}_r(\bar{x}).$$

Similarly, we can use $|\bar{x}_{i_r^+(\bar{x})}| \le r$ and $j > i_r^+(\bar{x})$ to conclude that

$$0 - 4r \cdot j \le x_{i_r^+(\bar{x})} + r - 4r \cdot j \le x_{i_r^+(\bar{x})} - 4r \cdot i_r^+(\bar{x}),$$

which means that $f_{N,r}(x) = \max_{i \in [i_r^+(\bar{x})]}\{x_i - 4r \cdot i\} = f_{N,r}(x_1, \ldots, x_{i_r^+(\bar{x})}, 0, \ldots, 0)$, giving the robust zero-chain property.

The second property is well-known [see, e.g., 9], but we show it here for completeness. Consider the point $\tilde{x} = -\frac{R}{\sqrt{N}}\mathbf{1} \in \mathcal{B}_R(0)$. Clearly, $\inf_{z \in \mathcal{B}_R(0)} f_{N,r}(z) \le f_{N,r}(\tilde{x}) = -\frac{R}{\sqrt{N}} - 4r$. Moreover, for any $x$ with $i_r^+(x) \le N$ we have $f_{N,r}(x) \ge x_N - 4Nr \ge -(4N+1)r$. Combining these two bounds yields $f_{N,r}(x) - \inf_{z \in \mathcal{B}_R(0)} f_{N,r}(z) \ge \frac{R}{\sqrt{N}} - (4N-3)r \ge \frac{R}{\sqrt{N}} - 4Nr$ as required.

The final property follows from the fact that maximization preserves convexity and Lipschitz constants. $\qquad \square$

Lemma 56.1 is the main technical novelty of the section, while the other parts are known and stated for completeness. Combining Lemmas 55 and 56 with appropriate choices of $N$ and $d$ immediately gives the lower bound.

**Proposition 57.** *Let $\frac{r}{R}, \delta \in (0,1)$ and $d = \lceil 60(\frac{R}{r})^2 \log \frac{R}{\delta \cdot r} \rceil$. There exists a distribution $P$ over convex and 1-Lipschitz functions from $\mathcal{B}_R(0) \to \mathbb{R}$ and corresponding $r$-local oracles such that the following holds for any $r$-local algorithm. With probability at least $1 - \delta$ over the draw of $(f, \mathcal{O}) \sim P$, when the algorithm interacts with $\mathcal{O}$, its first $\lceil \frac{1}{10}(\frac{R}{r})^{2/3} \rceil$ queries are at least $R^{2/3}r^{1/3}$ suboptimal for $f$.*

*Proof.* Set $N = \lfloor \frac{1}{10}(\frac{R}{r})^{2/3} \rfloor$ and $d \ge \lceil \frac{60R^2}{r^2} \log \frac{R}{\delta r} \rceil \ge \lceil N + \frac{20R^2}{r^2} \log \frac{20NR^2}{\delta r^2} \rceil$. Apply Lemma 55 with Lemma 56.1 to argue that for any algorithm, with probability at least $1 - \delta$ the first $N$ queries $x_1, \ldots, x_N$ satisfy $i_r^+(\mathbf{U}^\top x_i) \le N$, and substitute into Lemma 56.2 to conclude that the suboptimality of each query is at least $(\sqrt{10} - \frac{4}{10})(R^2 r)^{1/3} \ge (R^2 r)^{1/3}$. $\qquad \square$

Proposition 57 shows that as long as we wish to solve the minimization problem to accuracy $\epsilon = o(R^{2/3}r^{1/3})$, for any $r$-local algorithm, there is a function requiring $\Omega((R/r)^{2/3})$ queries to an $r$-local oracle, which gives strictly more information than a ball optimization oracle, proving our desired lower bound, and consequently Theorem 20 follows as an immediate corollary. However, our acceleration scheme (Theorem 6) assumes unconstrained, smooth and strictly convex problems. We now outline modifications to the construction (34) extending it to this regime.

**Unconstrained domain.** Following the approach of Diakonikolas and Guzmán [17], we note that the construction $f(x) = \max\{\frac{1}{2}f_{N,r}(x), \|x\| - \frac{R}{2}\}$ provides a hard instance for algorithms with unbounded queries, because any query with norm larger than $R/2$ is uninformative about the rotation of coordinates and has a positive function value, so that the minimizer is still constrained to a ball of radius $R$.

**Smooth functions.** The smoothing argument of Guzmán and Nemirovski [20] shows that $f(x) = \inf_{x' \in \mathcal{B}_r(x)}\{f_{N,2r}(x') + \frac{1}{r}\|x' - x\|^2\}$ is an $r$-robust zero-chain that is also $2/r$-smooth and satisfies $|f(x) - f_{N,2r}(x)| \le r$ for all $x$. Consequently, the lower bound holds for $O(1/r)$ smooth functions.

**Strictly convex functions.** The function $f(x) = f_{N,r}(x) + \frac{r^{1/3}}{2R^{4/3}}\|x\|^2$ provides an $(r^{1/3}R^{-4/3})$-strongly convex hard instance, since we can add the strongly convex regularizer directly in the local oracle without revealing additional information, and the regularizer size is small enough so as not to significantly affect the optimality gap.

### G.1 Proof of Lemma 55

Recall the coordinate progress notation

$$i_r^+(x) \overset{\text{def}}{=} \min\{i \in [d] \mid |x_j| \le r \text{ for all } j \ge i\}.$$

Before giving the proof of Lemma 55, we remark that a number of papers [30, 12, 17, 9] contain proofs for variations of this claim featuring some differences between the types of oracles considered, which do not materially affect the argument. The proofs in these papers are distinct, and vary in the dimensionality they require. Our argument below uses a random orthogonal transformation similarly to Woodworth and Srebro [30], Carmon et al. [12], but uses a more careful union bound (35), similarly to that of Diakonikolas and Guzmán [17], which allows for a much shorter proof and also obtains tighter dimension bounds as in Diakonikolas and Guzmán [17], Bubeck et al. [9].

**Lemma 55.** *Let $\frac{r}{R}, \delta \in (0,1)$, $N \in \mathbb{N}$ and $d \geq \lceil N + \frac{20R^2}{r^2} \log \frac{20NR^2}{\delta r^2} \rceil$. Let $f : \mathcal{B}_R(0) \to \mathbb{R}$ be an $r$-robust zero-chain and let $\mathbf{U} \in \mathbb{R}^{d \times d}$ be a random orthogonal matrix and fix an $r$-local algorithm $\mathcal{A}$. With probability at least $1 - \delta$ over the draw of $\mathbf{U}$, there exists an $r$-local oracle $\mathcal{O}$ for $f_{\mathbf{U}}(x) \stackrel{\text{def}}{=} f(\mathbf{U}^\top x)$ such that the queries $x_1, x_2, \ldots$ of $\mathcal{A}$ interacting with $\mathcal{O}$ satisfy*

$$i_r^+(\mathbf{U}^\top x_i) \leq i \ \text{ for all } \ i \leq N.$$

*Proof.* Let $u_1, \ldots, u_d$ be the columns of $\mathbf{U}$. Definition 54 directly suggests an $r$-local oracle for $f_{\mathbf{U}}(x) = f(\mathbf{U}^\top x)$: at query point $\bar{x}$ the oracle returns $f_{\mathbf{U}}^{\bar{x}} : \mathbb{R}^d \to \mathbb{R}$ such that

$$f_{\mathbf{U}}^{\bar{x}}(x) = f(\langle u_1, x \rangle, \ldots, \langle u_{i_r^+(\mathbf{U}^\top \bar{x})}, x \rangle, 0, \ldots, 0).$$

The $r$-robust zero-chain definition implies that $\mathcal{O}(\bar{x}) = f_{\mathbf{U}}^{\bar{x}}$ is a valid response for an $r$-local oracle for $f_{\mathbf{U}}$. Moreover, the oracle answer to query $x_i$ only depends on the first $i_r^+(\mathbf{U}^\top x_i)$ columns of $u$. Define

$$p_i \stackrel{\text{def}}{=} \max_{j \leq i} i_r^+\left(\mathbf{U}^\top x_j\right)$$

to be the highest progress attained up to query $i$. With this notation, we wish to show that

$$\mathbb{P}\left(\bigcap_{i \leq N} \{p_i \leq i\}\right) \geq 1 - \delta.$$

Note that at round $i + 1$ the algorithm could query $x_{i+1} = R \cdot u_{p_i}$ which would satisfy $p_{i+1} = i_r^+(\mathbf{U}^\top x_{i+1}) = 1 + p_i$. Therefore, it is possible to choose queries so that $i_r^+(\mathbf{U}^\top x_i) = p_i = i$. However, any faster increase in $p_i$ is highly unlikely, because it would require attaining high inner product with a direction $u_j$ for $j > p_i$ about which we have very little information when $d$ is sufficiently large.

To make this intuition rigorous, we apply the union bound to the failure probability, giving

$$\mathbb{P}\left(\bigcup_{i \leq N} \{p_i > i\}\right) = \mathbb{P}\left(\bigcup_{i \leq N} \{p_i > i, \ p_{i-1} < i\}\right) \leq \sum_{i=1}^N \mathbb{P}(p_i > i, \ p_{i-1} < i), \qquad (35)$$

with $p_0 = 0$. We further upper bound each summand as

$$\mathbb{P}(p_i > i, p_{i-1} < i) = \mathbb{P}\left(\bigcup_{j \geq i} \{|\langle u_j, x_i \rangle| > r, \ p_{i-1} < i\}\right)$$

$$\leq (d - i + 1) \cdot \mathbb{P}\left(|\langle u_i, x_i \rangle| > r, \ p_{i-1} < i\right), \qquad (36)$$

where the last step uses a union bound and the exchangeablility of $u_i, u_{i+1}, \ldots, u_d$ under the event $p_{i-1} < i$. Note that the event $p_{i-1} < i$ implies that that $x_i$ depends on $\mathbf{U}$ only through $\mathbf{U}^{(<i)} \stackrel{\text{def}}{=} u_1, \ldots, u_{i-1}$, as these vectors allow us to compute the oracle responses to queries $x_1, \ldots, x_{i-1}$.[6] Formally, we may write

$$x_i = a_i(\mathbf{U}^{(<i)}) 1\{p_{i-1} < i\} + \tilde{a}_i(\mathbf{U}) 1\{p_{i-1} \geq i\},$$

for two measurable functions $a_i : \mathbb{R}^{d \times (i-1)} \to \mathbb{R}^d$ and $\tilde{a}_i : \mathbb{R}^{d \times d} \to \mathbb{R}^d$. Consequently, we have

$$\mathbb{P}\left(|\langle u_i, x_i \rangle| > r, \ p_{i-1} < i\right) = \mathbb{P}\left(|\langle u_i, a_i(\mathbf{U}^{(<i)}) \rangle| > r, \ p_{i-1} < i\right)$$

$$\leq \mathbb{P}\left(|\langle u_i, a_i(\mathbf{U}^{(<i)}) \rangle| > r\right).$$

Conditional on $\mathbf{U}^{(<i)}$, the vector $u_i$ is uniformly distributed in the $(d - i + 1)$-dimensional space $\text{span}\{u_i, \ldots, u_d\}$. Therefore, standard concentration inequalities on the sphere [see 7, Lecture 8] give

$$\mathbb{P}\left(|\langle u_i, a_i(\mathbf{U}^{(<i)}) \rangle| > r \mid \mathbf{U}^{(<i)}\right) \leq 2 \exp\left\{-\frac{r^2}{2\|a_i(\mathbf{U}^{(<i)})\|^2} \cdot (d - i + 1)\right\} \leq \frac{\delta}{d^2},$$

where in the final step we substituted $\|a_i(\mathbf{U}^{(<i)})\| \le R$, and our setting of $d$, which implies

$$d - i + 1 \ge d - N \ge \frac{20R^2}{r^2} \log \frac{20NR^2}{\delta \cdot r^2} \ge \frac{2R^2}{r^2} \log \frac{2d^2}{\delta}.$$

Substituting $\mathbb{P}(|\langle u_i, x_i \rangle| > r, \, p_{i-1} < i) \le \frac{\delta}{d^2}$ into the bounds (35) and (36) concludes the proof. $\square$