[Reviews · NeurIPS 2020]

Review 1

Summary and Contributions: This paper considers various optimization procedures for smooth convex optimization, based around what the authors call a “ball optimization oracle” for radius r: for some query point x, the oracle finds the minimizer of the function in an l_2 ball of radius r around x. If the global minimizer is l_2 distance R away from the starting point, then one should expect that roughly (R/r) calls to the oracle are enough to reach the global minimizer. As one of the main contributions of this work, the authors show that, surprisingly, one can actually achieve a better oracle complexity of roughly (R/r)^{2/3}, and they further establish lower bounds that show this rate is essentially tight. In addition, the authors show that under a certain Hessian stability condition, it is possible to implement the oracle using a type of Newton’s method. ============= UPDATE ============= I have read the authors’ response, and I remain convinced of the significance and value of this work, so I maintain my score.

Strengths: This paper significantly clarifies the power that may be drawn from the simply stated ball optimization oracle. While the unaccelerated variant is somewhat straightforward, the fact that it can be combined with a Monteiro-Svaiter-type acceleration scheme (leading to effectively “jumping around” unexplored regions) is immensely curious. Furthermore, the nearly matching lower bounds, and their techniques for showing them, may find use in subsequent lower bounding results, by carefully capturing the power of the oracle model at hand. In addition to the oracle-based results, the paper goes a step further to show how, for functions that exhibit a certain Hessian stability condition, it is possible to implement an approximate variant of the oracle. Interestingly, this condition is met by several important problems in machine learning including l infinity, lp, and logistic regression, improving upon or matching previous results. In doing so, this works helps to clarify the connection between higher-order smooth optimization and various instances of regression, and it all goes back to the very clean formulation of the deceptively simple ball optimization oracle. This paper is an important work which leads to fundamental improvements in our understanding of optimization theory on several levels, in addition to establishing explicit improvements over previous key results for well-studied problems in machine learning. Such would make an excellent contribution to the conference with significant value to the community, and so I recommend acceptance of this work.

Weaknesses: The work only nearly matches previous lp regression rates for p << infinity. Small comments: -In the introduction, it should be noted that the acceleration improvement happens for Euclidean norm-based ball optimization oracles.

Correctness: To the best of my understanding, yes.

Clarity: The paper is well written.

Relation to Prior Work: Relation to prior work is clearly discussed. Small comments: -Hessian stability has also been studied in the context of matrix scaling, where it was called “second-order robust”: Cohen, M. B., Madry, A., Tsipras, D., & Vladu, A. (2017). Matrix scaling and balancing via box constrained Newton's method and interior point methods. In 2017 IEEE 58th Annual Symposium on Foundations of Computer Science (FOCS) (pp. 902-913). IEEE. -The authors should be aware of the following result, which achieves a related epsilon dependence for l infinity regression: Ene, A., & Vladu, A. (2019). Improved Convergence for $\ell_1 $ and $\ell_∞ $ Regression via Iteratively Reweighted Least Squares. In International Conference on Machine Learning (pp. 1794-1801).

Reproducibility: Yes

Additional Feedback:


Review 2

Summary and Contributions: The paper presents an accelerated method with ball minimization oracle which has better theoretical complexity than existing ones. The considered oracle seems to be quite artificial at first glance, but the authors demonstrate several applications where the new algorithm give better complexity for optimization problems in ML. In particular, the authors obtain better than state-of-the-art complexity for logistic regression and l_\infty regression, and comparable to existing bounds for general l_p regression. A lower complexity bound is also presented for algorithms with ball minimization oracle, yet under weaker assumptions than the ones used for the analysis of the proposed algorithm. ======================= After rebuttal I would like to thank the authors for their response. They agreed with my point on theory vs practice. So, I would like to keep the score unchanged.

Strengths: The paper presents an interesting framework for accelerating under ball minimization oracle which has several important ML applications. The mathematics behind is well elaborated and is a very good combination of different blocks with careful taking care of the details. Moreover, the theory has important implications for the question of complexity of standard ML problems both in terms of upper and lower bounds. Overall a significant and novel theoretical contribution for relevant ML problems.

Weaknesses: The framework is very theoretical and no experimental evidence is presented in the paper. Moreover, the proposed algorithm has several levels of inner-outer loops and my experience says that in practice algorithms with only two nested cycles and better complexity bounds work slower than simple algorithms with worse bounds.

Correctness: As far as I see, the proofs for the upper bounds are correct. I could not check the proofs for the lower bounds since I am not a good specialist in such questions.

Clarity: Overall the paper is clearly written.

Relation to Prior Work: It seems that a comparison with https://papers.nips.cc/paper/8980-globally-convergent-newton-methods-for-ill-conditioned-generalized-self-concordant-losses.pdf is missing. Concerning inexactly finding the iterate z in the accelerated gradient method, there are the following references which may be relevant https://www2.isye.gatech.edu/~nemirovs/LMCOLN2020WithSol.pdf Sect 5.5.1.2 https://epubs.siam.org/doi/abs/10.1137/140992382 https://arxiv.org/abs/2001.09013

Reproducibility: Yes

Additional Feedback: line 753. $t$ should be under the square root, as far as I understand.


Review 3

Summary and Contributions: This paper deals with the question of optimization of a smooth convex function, when given access to a "ball optimization oracle" that can find the optimizer of the function restricted to a ball of radius r centered at an input point x. Given access to such an oracle, it is straightforward to find the optimizer in roughly R/r calls to the oracle, where R is the initial distance from the optimum. This paper shows that it is possible to find the optimizer in roughly (R/r)^(2/3) calls to the oracle, and complements this result with a lower bound of the same order. ========= UPDATE: I have read the rebuttal, and I maintain my support for this paper.

Strengths: The paper builds on the Montero-Svaiter acceleration framework (and recent works that build on this approach for acceleration in highly-smooth functions), and shows the somewhat surprising result that acceleration can work in such a general abstraction. Nesterov's classic accelerated gradient methods and numerous follow up works have had a tremendous impact on the optimization community, and is a staple tool in machine learning optimization today. This work adds to our current understanding of acceleration. The paper gives several interesting applications of their framework, giving faster algorithms for logistic regression, l-inf regression, and comparable to state-of-the-art algorithms for L-p regression.

Weaknesses: The paper does not make an empirical evaluation of their results, but regardless, I think this is a very strong submission.

Correctness: I haven't checked the claims in detail, but I believe the claims in the paper.

Clarity: - Please explicitly write that the norm ball used in the introduction corresponds to the 2 norm. It's important because later the work also deals with l_p norms for other p. - The notation around equations 1-2 can be clearer. What are lambda_r(z) and y(lambda)?

Relation to Prior Work: - For lp optimization for the special case of p=4, the work of Bullins '19 gives a faster algorithm, and I think it's appropriate to cite it. [Disclaimer: I am not the author of that work]

Reproducibility: Yes

Additional Feedback: - The broader impact section is actually the conclusion of the paper. Since this work is entirely theoretical and does not address new questions, I do not foresee any significant ethical questions arising as a result of this work. Still, the authors should discuss this in the section appropriately.


Review 4

Summary and Contributions: UPDATED: Thanks a lot for your answer! I am happy to increase my score. ================================ In this paper, the authors propose a new oracle model for unconstrained convex minimization, which computes the minimum of the objective over the ball of fixed radius around a given point. Using this oracle model, they show how to minimize the function on the whole space, with the number of iterations proportional to R/r up to logarithmic terms, where R is the initial distance to the solution, and r is the radius of the oracle ball. Further, they show that this rate can be accelerated by using the Monteiro-Svaiter acceleration framework (which is well-known for obtaining optimal rates for high-order methods). It turns out, that logarithmic number of the new oracle calls can approximate well one (inexact) proximal step of Monteiro and Svaiter. The resulting complexity is (R/r)^{2/3}, up to logarithmic terms. The authors also show that this complexity bound is optimal. Further, they provide an example of the case, when the ball minimization oracle can be efficiently implemented, by doing second-order approximation of the objective and solving the corresponding trust-region subproblem (minimization of the quadratic function over the ball) by the Fast Gradient Method. For the functions whose Hessian is 'stable', they justify fast linear convergence for this subroutine. This improves the known complexity estimates for this problem class (taking the square root of condition number; the condition number is the constant of Hessian stability). Finally, they consider some applications of their approach for solving different logistic and l_p-regression problems.

Strengths: The results of the paper are theoretically sound and novel, to the best of my knowledge. The main contribution is the new oracle model for convex optimization and its analysis with the accelerated framework of Monteiro and Svaiter. It fits well with some recent advances in high-order methods (in particular, the results from [9] and from "Yurii Nesterov. Inexact accelerated high-order proximal-point methods. CORE DP 2020/08"), augmenting and extending the proximal-point approach. I believe, that this is a significant theoretical contribution, with a positive impact for optimization community.

Weaknesses: 1. My main concern would be the practical usage of the proposed approach. Despite the fact that the theory looks very nice, the benefit of using the multilevel optimization scheme remains somewhat questionable to me. It would be really interesting to see some numerical examples. Probably, this work would be much more suitable for optimization venue, than for NeurIPS conference. 2. Do I understand right, that for mu-strongly convex functions with L-Lipschitz continuous gradients, the constant of Hessian stability 'c' is equal to 'L / mu'? Therefore, for this problem class, Theorem 8 (and the corresponding Algrorithm 7) does not improve upon the classical Gradient Method (and thus loose to the basic Fast Gradient Method). Overall, this is not clear, do we obtain any gain from this framework for solving some optimization problems from the standard problem classes.

Correctness: Most of the claims looks correct. Though, I was not able to check all the proofs.

Clarity: The paper is generally well written. Statement of Theorem 20: it is not completely clear, what is 'algorithm's coin flips'? It would be also helpful to formally specify (or provide sufficient references), what is the 'distribution over convex functions' -- is it a random process?

Relation to Prior Work: There is a quite extensive discussion of the related work in Section 1.2. Personally, I would be interested to see an extra discussion, comparing the complexities of the new methods with already known bounds (especially, for problems from from Section 4, Applications).

Reproducibility: Yes

Additional Feedback:

[Author Response · NeurIPS 2020]

We thank the reviewers for their thoughtful comments and helpful feedback. Below, we address each reviewer in turn.

**Review 1.** $\ell_p$ *regression rates.* We agree that the $\ell_p$ regression runtimes obtained through our framework do not match the performance achieved by the state-the-art algorithms designed for the problem. However, we achieve these runtimes through a general framework that also yields state-of-the-art runtimes for $\ell_\infty$ and logistic regression. We believe that this sheds further light on the problem and is a step towards a more complete understanding of the problem class.

*Other comments.* In the revised version of the paper, we will emphasize in the intro that we obtain acceleration when our ball optimization oracle is restricted to affine transformations of Euclidean space (as opposed to a different norm, e.g., $\ell_1$ or $\ell_\infty$). We will also add further comparison to notions of Hessian stability in different norms appearing in the literature; for instance, the work of Cohen et al. solves ball-optimization problems where the ball is measured in the $\ell_\infty$ norm. Regarding Ene and Vladu's IRLS-based algorithm for $\ell_\infty$, thank you very much for pointing this out. We will add a comparison to this result in our revision. Our algorithm works with a slightly different treatment of the constraint matrix $\min_x \|\mathbf{A}x - b\|_\infty$ (versus $\min_{\mathbf{A}x=b} \|x\|_\infty$), bounds iteration complexity in terms of $\|x^*\|_2$ (versus $\sqrt{m}\|x^*\|_\infty$ where $m$ as dimension of $x^*$), and minimizes a softmax objective (versus $\ell_\infty$); we nevertheless believe our results can be adapted to their setting and our complexity is no worse (up to logarithmic factors) than theirs in the worst case and will comment on this in the revision.

**Reviewer 2.** *Theory vs. practice.* We agree that the main contribution of the paper lies on the theoretical side; our work provides a conceptually simple algorithmic framework with proof-of-concept complexity guarantees for a range of fundamental ML problems. Our proposed algorithms are not immediately suited for implementation, but opens the door to a promising new direction for designing practical algorithms with fewer loops and line searches.

*Additional references.* We appreciate the pointers to relevant papers. Using the notation of our paper, [FBR19] attains runtime bounds proportional to $MR$ (which is at most $\sqrt{L/\mu}$), where $M$ is the quasi-self-concordance (QSC) parameter and $R$ is the domain size; our paper attains improved runtime guarantees proportional to $(MR)^{2/3}$. Apart from that, the runtimes of both papers depend polylogarithmically on the problem condition number $L/\mu$. We will be sure to include this discussion in detail and the three relevant references on approximate solvers for AGD in the revision.

**Reviewer 3.** *Norm notation.* We use the general norm notation to include norms induced by any PSD matrix $\mathbf{M}$, which covers Euclidean $\ell_2$ as a special case when $\mathbf{M} = \mathbf{I}$. This general norm is used throughout the paper, e.g. Def. 1, Cor. 12, and particularly for applications where $\mathbf{M} = \mathbf{A}^\top \mathbf{A}$. We will be sure to clarify this generality when using the notation.

*Eqs. (1)–(2) notation.* For ball optimization oracle with center $z$, radius $r$, the value of $\lambda$ is a function of $r, z$ which we denote by $\lambda_r(z)$; $y(\lambda)$ is the unique point $y$ prescribed by a value $\lambda$ given $x, v$. Thus, the implementation of the MS oracle boils down to solving the implicit equation $\lambda = \lambda_r(y(\lambda))$. We will make this clearer in the revision.

*Bullins' paper.* Thank you for pointing this out. While Bullins' paper does give the state-of-the-art algorithm for $\ell_p$ regression when $p = 4$, a more recent work by [AKPS19] (ref. [2] of our paper) achieves the state of the art polynomial dependence for $\ell_p$ regression for all $p \geq 2$ and matches Bullins' work in the special $p = 4$ case. We will be sure to discuss Bullins' result in our revision.

*Broader impact.* Thank you for the suggestion; we will include more discussion in the revision.

**Reviewer 4.** *Practicality and relevance to NeurIPS.* Our paper is theoretical, but we hope that our conceptual insights—particularly the ball optimization abstraction and its connection to Hessian stability—may inspire practical developments. We believe the paper is relevant to the ML audience because the problems for which we prove faster runtime bounds include soft-margin SVM and logistic regression, which are central in ML; therefore, many of the researchers likely to turn our theoretical insight into practice belong to the NeurIPS community. On a related note, we remark that many of the recent (purely theoretical) developments of MS acceleration appeared in ML venues (see refs. [9,17] in our paper and the Bullins (2020) paper pointed out by Reviewer 3).

*Discussion on Theorem 8.* It is true that $\mu$-strongly convex $L$-smooth functions are $L/\mu$-Hessian stable globally (over a ball of radius $r = \infty$). However, for a ball of radius $r < \infty$, the problems that we consider in our applications section have a stability parameter $c$ much smaller than their condition number. Hessian stability is thus a distinct structural property from smoothness and strong convexity, allowing efficient minimization of certain poorly-conditioned functions.

*Statement of Theorem 20.* "Coin flips of an algorithm" is a common term for internal randomization in the algorithm. We explicitly construct the distribution over convex functions Appendix G by composing the the zero-chain defined in Eq. (34) with random orthogonal transformations. We will be sure to clarify this part in the revision.

*Runtime comparisons for applications.* For $\ell_p$ regression, we included known complexities from existing works in lines 109–122. For logistic and $\ell_\infty$ regression, the runtime comparisons are currently discussed in lines 66–85, and in more detail in the last paragraph of each application in Section 4. We will offer a more detailed comparison in the revision.

[Meta-Review · NeurIPS 2020]

This paper is concerned with optimization via a "ball optimization oracle", which returns the minimizer of a function restricted to an L2 ball of radius r around a query point x. The authors demonstrate an oracle complexity of roughly (R/r)^{2/3} when combined with a Monteiro-Svaiter acceleration scheme. The authors show that this oracle can be implemented on a variety of important machine learning problems. (Although, the authors acknowledge in the response that the algorithm is not immediately suitable for use.) The ideas in this paper are elegant and surprising, despite arising from a "deceptively simple" oracle. The reviewers were unanimously positive about this work, and everyone agrees it is an important theoretical contribution to the optimization community. I am delighted to recommend acceptance. Please address all of the reviewer comments. In particular, please revise the Broader Impact section to comport with the instructions in the Call for Papers.